applied mathematics/fluid mechanics

Carreau fluid, Womersley number, oscillatory channel flow

**Author for correspondence:**
S. Tabakova
e-mail: stabakova@gmail.com

# Oscillatory Carreau flows in straight channels

## S. Tabakova[1], N. Kutev[2] and St. Radev[1]

[1]Institute of Mechanics, and [2]Institute of Mathematics and Informatics, Bulgarian Academy of Sciences, Sofia, Bulgaria

 ST, 0000-0003-2271-9246

The present paper studies the oscillatory flow of Carreau fluid in a channel at different Womersley and Carreau numbers. At high and low Womersley numbers, asymptotic expansions in small parameters, connected with the Womersley number, are developed. For the intermediate Womersley numbers, theoretical bounds for the velocity solution and its gradient, depending on the problem parameters, are proven and explicitly given. It is shown that the Carreau number changes the type of the flow velocity to be closer to the Newtonian velocity corresponding to low or high shear or to have a transitional character between both Newtonian velocities. Some numerical examples for the velocity at different Carreau and Womersley numbers are presented for illustration with respect to the similar Newtonian flow velocity.

## 1. Introduction

The non-Newtonian character of blood and other viscoelastic fluids, e.g. polymers, important for some chemical and biochemical engineering applications, are usually described by generalized models of the Newtonian fluids [1]. These models assume the fluids as incompressible and propose a nonlinear dependence of the shear stress on the shear rate, such that the viscosity, which is a constant for the Newtonian fluids, to become a function of the shear rate. For different types of non-Newtonian fluids, this function is empirically determined and represents the rheological model of the fluid. For pseudoplastic or shear-thinning fluids, whose viscosity decreases with the shear rate, the model function is usually a power function (power-law model) or a rational function of the shear rate (Cross model, Carreau model, etc.) [2,3].

The shear-thinning viscosity, for example of blood, is quite well approximated by the Carreau viscosity model, as it has two Newtonian plateaus of constant viscosity at low and high shear rates. These plateaus are connected with a power-law region for the intermediate shear rates. For fluids whose viscosity is described by the Carreau model (Carreau fluids), the dimensionless parameter, Carreau number, is appropriate to be used, which is defined as the ratio between the characteristic shear

**Figure 1.** Sketch of the different flow regimes in the parameter-space diagram of Womersley number $\beta$ and Carreau number $Cu$.

rate and the transition shear rate [2,4,5]. It is assumed that for low values of the Carreau number the fluid behaviour is localized in the upper Newtonian plateau (low shear rate), while for its higher values the fluid behaviour is essentially defined by the Newtonian flow at the lower viscosity (as shown in figure 1). Usually, in these limiting cases, the flow velocity in straight pipes or channels is similar to that obtained with the Newtonian model based on the higher or lower viscosity. However, this general observation is approximative and depends on the other parameters of the problem. For example, in [6,7], it is shown through simulations in large arteries, that the blood flow, approximated as a Newtonian flow, can lead to errors, particularly in the presence of secondary flows due to curvatures.

Since the blood flow is pulsatile, the problems become additionally complicated if the non-Newtonian character is taken into account [7–9]. Apart from the Reynolds number, an additional parameter is introduced for the pulsatile or oscillatory flows, the so-called Womersley number (expressing the ratio of the local pulsating force to the viscous one) [10–12]. At low values of the Womersley number, both the Newtonian and Carreau flows in pipes or channels correspond to Poiseuille velocity profiles, while at the high values, they correspond to boundary layers on the walls (usually named Womersley or inertia flow regime). The position of the Poiseuille flow, Womersley flow and flow between these two regime flows is given in figure 1 as a parameter-space diagram of Carreau number and Womersley number. Also, the diagram contains the position of low shear, corresponding to high viscosity, and high shear–low viscosity. Thus, the combination of the Carreau and Womersley numbers can have an interesting influence on the flow characteristics for pulsatile non-Newtonian flows in straight pipe or channel, which is the purpose of the present work.

The pipe flow of Newtonian fluid due to oscillating pressure gradient has been first studied experimentally by Richardson & Tyler [13]. Their observations of the maximum velocity displacement towards the wall is known as the Richardson's annular effect, which is explained also using the analytical solution for the velocity [14,15]. The channel flow has similar behaviour with a slightly different analytical solution as found in [16] and later used by different authors to validate their numerical solutions for non-Newtonian fluid flows, e.g. [17–19].

In our previous studies [20–25], we have examined the problems of oscillating Carreau blood flow in a straight rigid channel or tube (non-deformable artery) and found numerical solutions for the flow velocity. Also, we have proven that the flow velocity and its gradient are limited from below and above by constants, which depend only on the lower value of the two Newtonian plateau viscosities, amplitude and frequency of the imposed oscillating pressure. Based on these results, we may assume that the blood flow is sufficiently exactly approximated by the Newtonian flow (based on the lower viscosity value) in the larger blood vessels, for example, the abdominal aorta, while in the smaller vessels, like the carotid artery, the non-Newtonian flow character is essential, which is also experimentally accepted [26].

The present paper aims to study more general cases of flows at different Womersley and Carreau numbers and to give solution estimates of the velocity and its gradient for a Carreau flow with respect to the similar Newtonian flow. The high and low Womersley number cases will be studied

separately as asymptotic expansions in small parameters, connected with the Womersley number. For the intermediate Womersley numbers, when only numerical solutions can be found, theoretical bounds will be proposed. For their derivation, the comparison principle between sub- and supersolutions of nonlinear uniformly parabolic equations will be used. By means of suitably chosen barrier functions, *a priori* bounds for the velocity solutions and their gradients, depending on the problem parameters, will be proven and explicitly given. The proven bounds for the Carreau velocity, its gradients and the bounds for the absolute difference between the Newtonian and Carreau velocity solutions will be shown to be valid for every Womersley number, Carreau number and rheological power coefficient $n$. However, the bound for the absolute difference between the Newtonian and Carreau velocity solutions will be more useful at low values of Carreau number or in the limit $n \to 1$. At high Womersley numbers, it will be shown that the effective Carreau number is responsible for solution type, i.e. if the solution can be approximated with one or the other Newtonian velocity corresponding to low or high shear viscosity or will have a transitional character.

The paper is constructed as follows. The theoretical assumptions together with the dimensional analysis are firstly presented in §2. Then, in §3, the Newtonian velocity solutions at different Womersley numbers are given. Special attention to the case of intermediate Womersley numbers for Carreau flow is paid in §4, which is the basic theoretical part concerning the bounds of the solutions. Section 5 deals with the two special cases of high and low Womersley numbers with approximations of the Carreau velocity. Numerical results for different Womersley and Carreau numbers are illustrated by plots in §6. In §7, the obtained theoretical results are discussed together with some numerical examples. The conclusions are briefly stated in §8.

# 2. Theoretical assumptions

## 2.1. Shear thinning

There exist different rheological models for non-Newtonian fluids describing the rheology of such fluids, i.e. describing the relation between the stresses and shear rates. It occurs that the Carreau model [3,23,27] is one of the most appropriate models for the so-called shear-thinning fluids: fluids whose viscosities gradually decrease with the increase of the angular deformation rate (shear rate). Moreover, the viscosities reach two limiting values, in the form of two different plateaus, corresponding to the higher, $\mu_0$, and lower viscosity, $\mu_\infty$, obtained at the low and high shear rate $\dot{\gamma}$, respectively. Apart from these two viscosities, the Carreau model also includes the characteristic time $\lambda$, which is equal to the inverse of the transition shear rate $\dot{\gamma}_t$, $\lambda = 1/\dot{\gamma}_t$, and the power coefficient $n$, where $0 < n < 1$. Then the Carreau viscosity $\mu_c$ is given by

$$\mu_c(\dot{\gamma}) = \mu_\infty + (\mu_0 - \mu_\infty)[1 + \lambda^2 \dot{\gamma}^2]^{(n-1)/2}. \tag{2.1}$$

The values of the physical constants for some shear-thinning fluids, whose viscosity is approximated by equation (2.1), and further named as Carreau fluids, can be found in the literature [1,3,21]. Usually, the Carreau fluids are considered as incompressible at isothermal conditions, i.e. with constant density, $\rho$, that is assumed in our present work.

## 2.2. Oscillatory

A two-dimensional straight infinite channel ($-\infty < x < \infty$), with width equal to $H$ ($0 \le y \le H$), is considered. The flow is supposed as unsteady laminar, driven by an oscillatory pressure gradient $A \cos \omega t$ along the channel axis $Ox$, with pulse amplitude $A$ and angular frequency $\omega$, such that $\partial p / \partial x = -A \cos \omega t$, where $p$ is the pressure.

The equations of continuity and motion in vector form are

$$\nabla \cdot \mathbf{v} = 0 \tag{2.2}$$

and

$$\rho \left( \frac{\partial \mathbf{v}}{\partial t} + \mathbf{v} \cdot \nabla \mathbf{v} \right) = -\nabla p + \nabla \cdot \mathbf{T}, \tag{2.3}$$

where $\mathbf{v} = (v_x, v_y)$ is the velocity vector, $\mathbf{T} = 2\mu_c(\dot{\gamma})\mathbf{E}$ is the viscous stress tensor, $\mathbf{E}$ is the shear rate tensor and $\dot{\gamma}$ is the shear rate: $\dot{\gamma}^2 = 2tr(\mathbf{E}^2)$.

From the assumption of an infinite channel and from equations (2.2) and (2.3), it follows that $\partial v_x / \partial x = 0$, $v_y = 0$ and $\partial p / \partial y = 0$. The only non-zero terms of $\mathbf{T}$ are $\tau_{xy} = \tau_{yx} = \mu_c(\dot{\gamma})\dot{\gamma}$, where $\dot{\gamma} = \partial v_x / \partial y$ is the

shear rate. Then equations (2.2) and (2.3) reduce to one single equation for $v_x$

$$\rho \frac{\partial v_x}{\partial t} = A \cos \omega t + \frac{\partial}{\partial y}\left(\mu_c . \frac{\partial v_x}{\partial y}\right), \tag{2.4}$$

where $\mu_c = \mu_c(\partial v_x/\partial y)$, with no-slip boundary conditions along the channel walls, i.e. $v_x = 0$ at both $y = 0$ and $y = H$.

## 2.3. Scaling analysis and dimensionless groups

Using $H$ as a characteristic length ($y = HY$), $1/\omega$ as a characteristic time ($t = T/\omega$) and $B$ as a characteristic velocity ($v_x = BU$), the dimensionless form of equation (2.4), together with (2.1), becomes

$$8\beta^2 \frac{\partial U}{\partial T} - \frac{\partial}{\partial Y}\left\{\left[1 - c + c\left(1 + Cu^2\left(\frac{\partial U}{\partial Y}\right)^2\right)^{(n-1)/2}\right]\frac{\partial U}{\partial Y}\right\} - \cos(T) = 0, \tag{2.5}$$

where $B = A\,H^2/\mu_0$, $\beta = (H/2)\sqrt{\rho\omega/2\mu_0}$—the Womersley number [11], $n \in (0, 1)$, $c = 1 - (\mu_\infty/\mu_0)$ ($1 \geq c > 0$), $\mu_0$—the characteristic viscosity and $Cu = \lambda B/H$—the Carreau number [2]. The dimensionless no-slip boundary conditions on the channel walls are

$$U(0, T) = U(1, T) = 0. \tag{2.6}$$

The introduction of the Carreau number is appropriate to distinguish the different cases of shear-thinning: Newtonian flow with the higher viscosity at $Cu = 0$; low shear thinning at $Cu \ll 1$; medium shear thinning at $Cu \sim O(1)$; high shear thinning at $Cu \gg 1$. In the next sections, we shall discuss the influence of $Cu$ on the velocity solution for the different cases with respect to $\beta$, namely three different cases of $\beta \to 0$, $\beta \sim O(1)$ and $\beta \to \infty$.

# 3. Solutions of the Newtonian flow velocity for different Womersley number cases

From equation (2.5), it is clear that the Womersley number $\beta$, significantly changes this equation. For low $\beta$, the velocity profiles are Poiseuille like, while for high $\beta$, in the limit $\beta \to \infty$, they are with boundary layer character, but still symmetric with respect to $Y = 1/2$.

As mentioned above, at $Cu = 0$, the fluid is Newtonian. From equations (2.5) and (2.6), its velocity, further denoted by $V(T, Y)$, is found explicitly for the first time in the classical book of Landau & Lifshitz [16] and also used in our previous papers [20–25]

$$V(T, Y) = \frac{1}{8\beta^2}[E(Y, \beta)\sin T + D(Y, \beta)\cos T], \tag{3.1}$$

where

$$\left.\begin{aligned}
E(Y, \beta) &= 1 + \frac{1}{1 - \cos^2\beta - \cosh^2\beta}[S_1(Y, \beta)S_2(\beta) + C_1(Y, \beta)C_2(\beta)]\\
D(Y, \beta) &= \frac{1}{1 - \cos^2\beta - \cosh^2\beta}[S_1(Y, \beta)C_2(\beta) - C_1(Y, \beta)S_2(\beta)]
\end{aligned}\right\} \tag{3.2}$$

and

$$\left.\begin{aligned}
S_1(Y, \beta) &= \sin\left[2\beta\left(Y - \frac{1}{2}\right)\right]\sinh\left[2\beta\left(Y - \frac{1}{2}\right)\right], & S_2(\beta) &= \sin\beta\sinh\beta\\
C_1(Y, \beta) &= \cos\left[2\beta\left(Y - \frac{1}{2}\right)\right]\cosh\left[2\beta\left(Y - \frac{1}{2}\right)\right], & C_2(\beta) &= \cos\beta\cosh\beta.
\end{aligned}\right\} \tag{3.3}$$

If the lower viscosity $\mu_\infty$ is used as a characteristic viscosity and $B_\infty = \frac{AH^2}{\mu_\infty} = B/(1 - c)$ as characteristic velocity, the corresponding Newtonian flow velocity $f(T, Y)$ satisfies the equation

$$8\beta_\infty^2 \frac{\partial f}{\partial T} - \frac{\partial^2 f}{\partial Y^2} - \cos(T) = 0, \tag{3.4}$$

where $\beta_\infty = \beta\sqrt{\mu_0/\mu_\infty} = (\beta/\sqrt{1-c}) > \beta$. The solution $f(T, Y)$ has the same form as (3.1)–(3.3) at $\beta$ replaced by $\beta_\infty$ [21]. However, for many polymer fluids, $\mu_\infty$ is assumed as zero [1] and then its usage is limited.

In order to give some further estimates of the Carreau fluid velocity in connection with the Newtonian solution (3.1), the latter is developed into series of $\beta \to 0$ and denoted by $V_0(T, Y)$

$$V_0(T, Y) = V_{00}(T, Y) + \beta^2 V_{01}(T, Y) + O(\beta^4), \tag{3.5}$$

where

$$V_{00}(T, Y) = \frac{\cos T}{2}(Y - Y^2), \quad V_{01}(T, Y) = \frac{\sin T}{3}(Y^4 - 2Y^3 + Y). \tag{3.6}$$

The Newtonian solution $f(T, Y)$ (corresponding to $\mu_\infty$) at $\beta \to 0$ and if $\beta_\infty = \beta/\sqrt{1-c} \to 0$, can be developed also as an asymptotic expansion in $\beta_\infty^2$, similarly to (3.5):

$$f_0(T, Y) = f_{00}(T, Y) + \beta_\infty^2 f_{01}(T, Y) + O(\beta_\infty^4), \tag{3.7}$$

where $f_{00} = V_{00}$ and $f_{01} = V_{01}$.

In the limit $\beta \to \infty$, the solution (3.1) is developed into series of the small parameter $1/\beta^2$ and denoted by $V_\infty(T, Y)$

$$V_\infty(T, Y) = \frac{V_{\infty 0}}{8\beta^2} + O\left(\frac{1}{\beta^4}\right), \tag{3.8}$$

where

$$V_{\infty 0}(T, Y) = \sin T - \exp(-2\beta Y)\sin(T - 2\beta Y) - \exp(-2\beta(1 - Y))\sin(T - 2\beta(1 - Y)). \tag{3.9}$$

The solution above is found to be a uniformly valid solution, after applying the perturbation theory [28]. The first term corresponds to the solution in the interior region (between the two boundary layers), the second and third ones—to the solutions in the boundary layers near the walls $Y = 0$ and $Y = 1$, respectively.

Moreover, in the limit $\beta \to \infty$, also $\beta_\infty \to \infty$ and then the solution $f(T, Y)$ can be developed into series of $1/\beta_\infty^2$. It will have the same form as (3.8) and (3.9), but with $\beta$ replaced by $\beta_\infty$

$$f_\infty(T, Y) = \frac{f_{\infty 0}}{8\beta_\infty^2} + O\left(\frac{1}{\beta_\infty^4}\right). \tag{3.10}$$

In the general case of a Carreau fluid, at $Cu \neq 0$, the velocity satisfying equations (2.5) and (2.6), can be found only numerically. In the limiting case of low $Cu \to 0$, there exists an asymptotic solution, given in [22]. For $Cu/\beta \to \infty$, the solution of equations (2.5) and (2.6) is found numerically to be close to the solution $f(T, Y)/(1 - c)$. In the following section, we shall present some bounds of the solution $U(T, Y)$, its derivatives and its reference with respect to the Newtonian flow solution $V(T, Y)$.

# 4. Solution bounds at $\beta \sim O(1)$

## 4.1. Bounds for the Newtonian velocity and its derivatives

Below we shall list some bounds of the Newtonian solution (3.1), which will be further used to estimate its divergence from the Carreau solution. First, we turn back to equation (2.5) with boundary conditions (2.6) satisfied by $V(T, Y)$ (at $Cu = 0$)

$$L_0(V) = 8\beta^2 V_T - V_{YY} - \cos T = 0 \quad \text{for } T \in \mathbb{R}, \quad Y \in (0, 1) \tag{4.1}$$

and

$$V(T, 0) = V(T, 1) = 0 \quad \text{for } T \in \mathbb{R}, \tag{4.2}$$

where the derivatives with respect to $T$ and $Y$ are denoted as subscripts.

Some *a priori* bounds for the derivatives of $V(T, Y) : V_Y(T, Y)$ and $V_{YY}(T, Y)$ are proven in $\mathbb{R} \times [0, 1]$ in the next lemma (its proof is given in [21]). These bounds are valid for all values of $\beta$.

**Lemma 4.1.** *The gradient $V_Y(T, Y)$ of the Newtonian velocity solution attains its maximum and minimum on the boundary $\{(T, 0) \cup (T, 1); T \in \mathbb{R}\}$ and the bounds*

$$|V(T, Y)| \le \frac{1}{2} Y(1 - Y) \le \frac{1}{8}, \tag{4.3}$$

$$|V_Y(T, Y)| \le \frac{1}{2} B(\beta) \le \frac{1}{2} \tag{4.4}$$

*and*

$$|V_{YY}(T, Y)| \le 1, \tag{4.5}$$

*hold for every $T \in \mathbb{R}$ and $Y \in [0, 1]$, where*

$$B(\beta) = \frac{\sqrt{2}}{2\beta} \sqrt{\frac{\sinh^2 \beta + \sin^2 \beta}{\cos^2 \beta + \sinh^2 \beta}} \le 1. \tag{4.6}$$

## 4.2. *A priori* bounds for the Carreau flow

In this subsection, we shall prove some bounds for the Carreau fluid velocity and its gradients. For this purpose, equation (2.5) is rewritten in a non-divergence form in the following manner:

$$8\beta^2 U_T - [1 - c + c\Phi(U_Y)]U_{YY} = \cos T, \\ \text{for } T > 0, Y \in (0, 1) \tag{4.7}$$

with boundary and initial conditions

$$\left. \begin{array}{l} U(T, 0) = U(T, 1) = 0 \quad \text{for } T \ge 0 \\ U(0, Y) = V(0, Y) \quad \text{for } Y \in [0, 1] \end{array} \right\} \tag{4.8}$$

and

where

$$\Phi(\eta) = (1 + nCu^2\eta^2)(1 + Cu^2\eta^2)^{(n-3)/2}. \tag{4.9}$$

The case $c = 1$ will not be discussed in the present work. It seems more complicated, because equation (4.7) is not uniformly parabolic and the problem (4.7) and (4.8) has no more a global classical solution for $T \in [0, \infty)$ (cf. [29], where the discussed problem corresponds to ours at $n = 0$), since the gradient of the solution blows up on the boundary and the solution detaches from the boundary data after a finite time. However, the cases of $c \to 1$ are included in the present study.

In order to prove *a priori* bounds for the solution $U(T, Y)$ of the Carreau velocity satisfying (4.7)–(4.9), we need the following auxiliary result:

**Lemma 4.2.** *If $n \in (0, 1)$ then the function $\Phi(\eta) = (1 - n)(1 + Cu^2\eta^2)^{(n-3)/2} + n(1 + Cu^2\eta^2)^{(n-1)/2}$ is monotonically decreasing and satisfies the relations*

$$0 \le 1 - \Phi(\eta) \le \min\left\{1, \frac{3}{2}(1 - n)Cu^2\eta^2\right\} \quad \text{for } \eta \ge 0. \tag{4.10}$$

*Proof.* Since $\Phi(\eta) \ge 0$ and $0 < n < 1$, it follows that $0 \le 1 - \Phi(\eta) \le 1$. Tedious calculations give us

$$1 - \Phi(\eta) = (1 - n)[1 - (1 + Cu^2\eta^2)^{(n-3)/2}] + n[1 - (1 + Cu^2\eta^2)^{(n-1)/2}]$$

$$= -\frac{1}{2}(1 - n)(n - 3)Cu^2\eta^2 \int_0^1 (1 + Cu^2\theta\eta^2)^{(n-5)/2} \, d\theta$$

$$\quad - \frac{1}{2}n(n - 1)Cu^2\eta^2 \int_0^1 (1 + Cu^2\theta\eta^2)^{(n-3)/2} \, d\theta$$

$$= \frac{1}{2}(1 - n)Cu^2\eta^2 \int_0^1 [(3 - n)(1 + Cu^2\theta\eta^2)^{(n-5)/2} + n(1 + Cu^2\theta\eta^2)^{(n-3)/2}] d\theta$$

$$\le \frac{3}{2}(1 - n)Cu^2\eta^2,$$

which proves lemma 4.2. $\qquad \square$

On the basis of (4.10), the following theorems concerning the velocity gradient are proven.

**Theorem 4.3 (Global gradient bounds).** *Suppose $U(T, Y)$ is the solution of (4.7)–(4.9). Then $U_Y(T, Y)$ attains its maximum and minimum on the parabolic boundary $\Gamma = \{(T, 0) \cup (T, 1); T \geq 0\} \cup \{(0, Y); Y \in [0, 1]\}$ and the bound*

$$|U_Y(T, Y)| \leq \max\{\sup_{T \geq 0} |U_Y(T, 1)|, \sup_{T \geq 0} |U_Y(T, 0)|, \sup_{Y \in [0,1]} |V_Y(0, Y)|\} \tag{4.11}$$

*holds for $T \geq 0$, $Y \in [0, 1]$.*

*Proof.* Differentiating (4.7) and (4.8) with respect to $Y$, we get that $U_Y(T, Y)$ satisfies the boundary value problem

$$P(U_Y) = 8\beta^2(U_Y)_T - [1 - c + c\Phi(U_Y)](U_Y)_{YY}$$
$$+ Cu^2 c(1 - n)(3 + nCu^2 U_Y^2)(1 + Cu^2 U_Y^2)^{(n-5)/2} U_Y(U_Y)_Y = 0 \tag{4.12}$$

and $\qquad U_Y(0, Y) = V_Y(0, Y) \quad \text{for } Y \in [0, 1].$

From (4.10) and the regularity of $U(T, Y)$, it follows that the operator $P$ is uniformly parabolic. According to the strong maximum principle for uniformly parabolic equations (see Theorem 2, Section 3 in [30]) $U_Y(T, Y)$ attains its maximum and minimum on $\Gamma$ and the bound (4.11) holds. $\qquad\square$

**Theorem 4.4 (Boundary gradient bounds).** *Suppose $U(T, Y)$ is the solution of the Carreau flow problem (4.7)–(4.9). Then the bounds*

$$|U(T, Y)| \leq K_1^0 Y(1 - Y) \leq \frac{1}{4} K_1^0 \quad \text{for } T \geq 0, Y \in [0, 1] \tag{4.13}$$

*and*

$$|U_Y(T, 0)|, |U_Y(T, 1)| \leq K_1^0 \quad \text{for } T \geq 0 \tag{4.14}$$

*hold, where*

$$K_1^0 = \frac{1}{2(1 - c)}. \tag{4.15}$$

*Proof.* The function $H_1^0(T, Y) = K_1^0 Y(1 - Y)$ is a supersolution to the boundary value problem (4.7)–(4.9). Indeed, for the operator

$$L(Z) = 8\beta^2 Z_T - [1 - c + c\Phi(U_Y)]Z_{YY} \quad \text{in} \quad \mathbb{R}^+ \times (0, 1),$$

we have from the choice of $K_1^0$ and from (4.3)

$$L(H_1^0) = 2K_1^0[1 - c + c\Phi(U_Y)] \geq \cos T \quad \text{for } T > 0, Y \in [0, 1],$$

$$H_1^0(T, 0) = H_1^0(T, 1) = 0 \quad \text{for } T \geq 0$$

and $\qquad H_1^0(0, Y) = K_1^0 Y(1 - Y) \geq \frac{1}{2(1 - c)} Y(1 - Y) \geq \frac{1}{2} Y(1 - Y) \geq V(0, Y) \quad \text{for } Y \in [0, 1].$

From the comparison principle, we get

$$U(T, Y) \leq K_1^0 Y(1 - Y) \quad \text{for } T \geq 0, Y \in [0, 1].$$

Analogously, the function $-H_1^0(T, Y)$ is a subsolution to (4.7)–(4.9) and the opposite inequality

$$-K_1^0 Y(1 - Y) \leq U(T, Y) \quad \text{for } T \geq 0, Y \in [0, 1],$$

holds. The bounds (4.14) are consequence of (4.13). $\qquad\square$

From (4.11), (4.14) and (4.4), we get the following:

**Corollary 4.5.** *Suppose $U(T, Y)$ is the solution of the Carreau flow problem (4.7)–(4.9). Then the bound*

$$|U_Y(T, Y)| \leq K_1^0 = \frac{1}{2(1 - c)} \quad \text{holds for } T \geq 0, Y \in [0, 1]. \tag{4.16}$$

However, for $c \lesssim 1$ ($c$ close to 1, but still $c < 1$), the solution $U(T, Y)$ of (4.7) is bounded. This means that the constant $K_1^0$ given by (4.16) is not appropriate to estimate the gradient $|U_Y(T, Y)|$. In order to improve (4.16) for $c$ close to 1, we repeat iteratively the proofs of theorem 4.4 and corollary 4.5, starting with the initial iteration $K_1^0$.

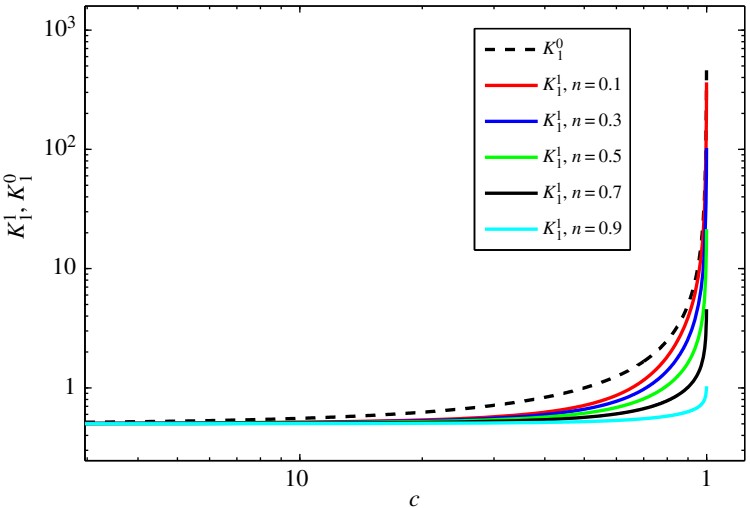

**Figure 2.** Plots of $K_1^0$ and $K_1^1$ as functions of $c$ for different $n$ from 0.1 to 0.9 by 0.2 and $Cu = 1$.

**Theorem 4.6 (Improved boundary gradient bounds).** *Suppose $U(T, Y)$ is the solution of (4.7)–(4.9).*
*Then the bounds*

$$|U(T, Y)| \leq K_1^1 Y(1 - Y) \leq \frac{1}{4} K_1^1 \quad \text{for } T \geq 0,\, Y \in [0, 1] \tag{4.17}$$

*and*

$$|U_Y(T, 0)|,\, |U_Y(T, 1)| \leq K_1^1 \quad \text{for } T \geq 0, \tag{4.18}$$

*hold, where*

$$K_1^1 = \frac{1}{2[1 - c + c\Phi(1/2(1 - c))]}. \tag{4.19}$$

*Proof.* Using (4.16), the function $H_1^1(T, Y) = K_1^1 Y(1 - Y)$ is a supersolution of (4.7)–(4.9), similarly to the proof of theorem 4.4. Hence from the comparison principle we get

$$U(T, Y) \leq K_1^1 Y(1 - Y) \quad \text{for } T \geq 0,\, Y \in [0, 1].$$

Analogously, the function $-H_1^1(T, Y)$ is a subsolution of (4.7)–(4.9) and the opposite inequality

$$-K_1^1 Y(1 - Y) \leq U(T, Y) \quad \text{for } T \geq 0,\, Y \in [0, 1]$$

holds, which proves (4.17). The bounds (4.18) are consequences of (4.17). □

From the expressions for $K_1^0$ and $K_1^1$ given by equations (4.15) and (4.19), it is obvious that $K_1^0 \geq K_1^1$ for all values of $c$, $Cu$ and $n$. Moreover, $K_1^1$ tends to $K_1^0$ at $n \to 0$ and/or $Cu \to \infty$. Here, we include a plot to show this tendency given in figure 2.

As a consequence of theorems 4.3 and 4.6, we get the following corollary:

**Corollary 4.7.** *Suppose $U(T, Y)$ is the solution of (4.7)–(4.9). Then the bound*

$$|U_Y(T, Y)| \leq K_1^1 = \frac{1}{2[1 - c + c\Phi(1/2(1 - c))]} \quad \text{holds for } T \geq 0,\, Y \in [0, 1]. \tag{4.20}$$

For $c \geq 0.5$, simple computations give the inequality

$$K_1^1 = \frac{1}{2[1 - c + c\Phi(1/2(1 - c))]} \leq \frac{1}{2n}(1 + Cu^2)^{(1-n)/2}\left(\frac{1}{2(1 - c)}\right)^{1-n}\left(1 + \frac{Cu^2}{4n^2}\right)^{(1-n)}, \tag{4.21}$$

the proof of which is presented in appendix A.

Repeating the proofs of theorem 4.6 and corollary 4.7, we get inductively the following bounds:

$$|U(T, Y)| \leq K_1^m Y(1 - Y) \quad \text{for } T \geq 0, \quad Y \in [0, 1], m = 2, 3, \ldots \tag{4.22}$$

$$|U_Y(T, 0)| \leq K_1^m, \quad |U_Y(T, 1)| \leq K_1^m \quad \text{for } T \geq 0, m = 2, 3, \ldots \tag{4.23}$$

$$|U_Y(T, Y)| \leq K_1^m, \tag{4.24}$$

where

$$K_1^m = \frac{1}{2[1 - c + c\Phi(K_1^{m-1})]}. \tag{4.25}$$

For $c \geq 0.5$ and $m \geq 2$, similarly to (4.21), the following inequality is inductively obtained

$$K_1^m \leq \frac{1}{2n} \left(\frac{1}{2(1-c)}\right)^{(1-n)^m} (1 + Cu^2)^{((1-n)^m)/2} \left(1 + \frac{Cu^2}{4n^2}\right)^{\sum_{j=1}^{m} (1-n)^j} \tag{4.26}$$

The proof of (4.26) is given in appendix B.

Since $\lim_{m\to\infty} \sum_{j=1}^{m} (1 - n)^j = (1 - n)/n$ and $\lim_{m\to\infty}(1 - n)^m = 0$, then after the limit $m \to \infty$ in (4.25) and (4.26), we get from (4.24) the bound

$$|U_Y(T, Y)| \leq K_1^\infty, \tag{4.27}$$

where

$$K_1^\infty = \frac{1}{2n} \left(1 + \frac{Cu^2}{4n^2}\right)^{(1-n)/n}. \tag{4.28}$$

Combining (4.20) and (4.27), the final bound is obtained

$$|U_Y(T, Y)| \leq K_1 \quad \text{for } T \geq 0, \quad Y \in [0, 1], \quad c > 0, \tag{4.29}$$

where

$$K_1 = \left\{ \begin{array}{ll} K_1^1 & \text{for } c \in (0, 0.5] \\ \min\{K_1^1, K_1^\infty\} & \text{for } c > 0.5 \end{array} \right\}. \tag{4.30}$$

Analogously, the bound for $U(T, Y)$ is derived

$$|U(T, Y)| \leq K_1^\infty Y(1 - Y) \leq \frac{1}{4} K_1^\infty \quad \text{for } T \geq 0, \quad Y \in [0, 1]. \tag{4.31}$$

Combining (4.17) and (4.31) the final bound is reached

$$|U(T, Y)| \leq K_1 Y(1 - Y) \leq \frac{1}{4} K_1 \quad \text{for } T \geq 0, \quad Y \in [0, 1] \quad \text{and} \quad c > 0. \tag{4.32}$$

## 4.3. Relations between the Carreau flow and Newtonian flow velocity

In this subsection, we shall give bounds of the difference between the Newtonian flow solution and the Carreau flow solution as well as of the difference between their gradients.

**Theorem 4.8.** *Suppose $U(T, Y)$ is the solution of the Carreau flow problem (4.7)–(4.9). Then the bounds*

$$|U(T, Y) - V(T, Y)| \leq K_2 Y(1 - Y) \leq \frac{1}{4} K_2 \quad \text{for } T \geq 0, Y \in [0, 1] \tag{4.33}$$

*and*

$$|U_Y(T, 0) - V_Y(T, 0)|, \quad |U_Y(T, 1) - V_Y(T, 1)| \leq K_2 \quad \text{for } T \geq 0 \tag{4.34}$$

*hold, where*

$$K_2 = (1 - n)Cu^2 \frac{3c}{4[1 - c + c\Phi(K_1)]} (K_1)^2 \tag{4.35}$$

*and $K_1$ is given by (4.30).*

*Proof.* The function $Z(T, Y) = U(T, Y) - V(T, Y)$ satisfies the boundary value problem

$$\left.\begin{aligned} L(Z) &= -c[1 - \Phi(U_Y)]V_{YY} \quad \text{in } \mathbb{R}^+ \times (0, 1), \\ Z(T, 0) &= Z(T, 1) = 0 \quad \text{for } T \geq 0 \\ Z(0, Y) &= 0 \quad \text{for } Y \in [0, 1], \end{aligned}\right\} \tag{4.36}$$

and

where the solution $V(T, Y)$ of the Newtonian flow is given by (3.1). From lemma 4.2, it follows that $\Phi(\eta)$ is a decreasing function of $\eta$. Then from (4.10) and (4.29), we obtain the inequality

$$\Phi(U_Y) \geq \Phi(K_1) \geq 1 - \frac{3}{2}(1 - n)Cu^2(K_1)^2. \tag{4.37}$$

The auxiliary function $H_2(T, Y) = K_2 Y(1 - Y)$ is a supersolution for (4.36). Indeed, simple computations give us from (4.5) and (4.37) the inequalities

$$L(Z) = -c[1 - \Phi(U_Y)]V_{YY} \leq c[1 - \Phi(U_Y)]|V_{YY}|$$

$$\leq c[1 - \Phi(K_1)]|V_{YY}| \leq \frac{3}{2}c(1 - n)Cu^2(K_1)^2$$

$$\leq 2K_2[1 - c + c\Phi(K_1)] \leq 2K_2[1 - c + c\Phi(U_Y)] = L(H_2) \quad \text{for } T \geq 0, Y \in (0, 1).$$

Since

$$H_2(T, 0) = H_2(T, 1) = 0 \quad \text{for } T \geq 0$$

and

$$H_2(0, Y) = K_2 Y(1 - Y) \geq 0 \quad \text{for } Y \in [0, 1],$$

and using the comparison principle, we get

$$U(T, Y) - V(T, Y) \leq K_2 Y(1 - Y) \leq \frac{1}{4}K_2 \quad \text{for } T \geq 0, Y \in [0, 1].$$

Analogously, by means of $-H_2(T, Y)$ the opposite bound

$$-K_2 Y(1 - Y) \leq U(T, Y) - V(T, Y)$$

holds. Bounds (4.34) are a consequence of (4.33). $\qquad \square$

# 5. Carreau flow velocity estimates at low and high $\beta$

## 5.1. $\beta \to 0$

In the limit $\beta \to 0$, similarly to the Newtonian solution (3.5), the Carreau fluid velocity $U_0(T, Y)$ is developed into series of $\beta^2$

$$U_0(T, Y) = U_{00}(T, Y) + \beta^2 U_{01}(T, Y) + O(\beta^4). \tag{5.1}$$

Then $U_{00}(T, Y)$ satisfies the uniformly elliptic equation for every fixed $T > 0$

$$\left.\begin{aligned} -\frac{\partial}{\partial Y}&\left\{\left[1 - c + c\left(1 + Cu^2\left(\frac{\partial U_{00}}{\partial Y}\right)^2\right)^{(n-1)/2}\right]\frac{\partial U_{00}}{\partial Y}\right\} = \cos(T), \\ &\text{for } T > 0, Y \in (0, 1) \\ U_{00}&(T, 0) = U_{00}(T, 1) = 0 \quad \text{for } T \geq 0, \end{aligned}\right\} \tag{5.2}$$

and

which is different from the uniformly parabolic equation (2.5).

The function $U_{00}$ is uniquely defined for $T \in \mathbb{R}$ and $Y \in (0, 1)$ as a classical solution of the boundary value problem (5.2). From the uniqueness theorem, it follows that $U_{00}(T, Y)$ is $2\pi$ – periodic function of $T$, symmetric in $Y$ for $Y = 1/2$ and $U_{00}(-(\pi/2) + k\pi, Y) \equiv 0$ for $Y \in [0, 1]$, where $k = \pm 1, \pm 2, \ldots$ If equation (5.2) is rewritten as (4.7) in the form

$$-\left[1 - c + c\Phi\left(\frac{\partial U_{00}}{\partial Y}\right)\right]\frac{\partial^2 U_{00}}{\partial Y^2} = \cos T, \tag{5.3}$$

it can easily be shown that $U_{00}(T, Y)$ is a concave function of $Y$ for $T \in (-(\pi/2), \pi/2)$ and a convex one for $T \in (\pi/2, 3\pi/2)$.

From equations (5.3) and (3.6), it follows that

$$-\frac{\partial^2 U_{00}}{\partial Y^2} \geq \cos T = -\frac{\partial^2 V_{00}}{\partial Y^2} \quad \text{for } T \in \left(-\frac{\pi}{2}, \frac{\pi}{2}\right), Y \in (0, 1)$$

and

$$-\frac{\partial^2 U_{00}}{\partial Y^2} \leq \cos T = -\frac{\partial^2 V_{00}}{\partial Y^2} \quad \text{for } T \in \left(\frac{\pi}{2}, \frac{3\pi}{2}\right), Y \in (0, 1).$$

Thus from the strong interior maximum principle for elliptic equations, we get

$$U_{00}(T, Y) > V_{00}(T, Y) > 0 \quad \text{for } T \in \left(-\frac{\pi}{2}, \frac{\pi}{2}\right), Y \in (0, 1)$$

and

$$U_{00}(T, Y) < V_{00}(T, Y) < 0 \quad \text{for } T \in \left(\frac{\pi}{2}, \frac{3\pi}{2}\right), Y \in (0, 1).$$

Since in the *a priori* bounds for $U(T, Y)$ the barrier function is independent of $T$ (similar to the one in the proof of theorem 4.8), the same bounds hold for $U_{00}(T, Y)$. Thus (4.33) holds also for $U_{00}(T, Y)$

$$|U_{00}(T, Y) - V_{00}(T, Y)| \leq K_2 Y(1 - Y) \leq \frac{1}{4} K_2 \quad \text{for } Y \in [0, 1]. \tag{5.4}$$

It is seen from equation (5.2) that the solution $U_{00}$ strongly depends on $Cu$. If $Cu \to 0$, since $(1 - n) < 1$, $K_2 \to 0$ and the solution $U_{00}$ is the Newtonian solution $V_{00}$, given with equation (3.6).

If, however, $Cu \gg 1$ and $c < 1$, such that $\beta_\infty \to 0$, the solution $U_{00}$ is close to the Newtonian solution, which corresponds to the lower viscosity $\mu_\infty$, i.e. to $f_{00}/(1 - c) = (\cos T/2(1 - c))(Y - Y^2)$.

Here, we have to note that the coefficient $1/(1 - c)$ appears because of the relation between the characteristic velocities and because the Carreau dimensional velocity is the same at both characteristic velocities, i.e. $BU \equiv B_\infty u$, where $u$ is the dimensionless Carreau velocity calculated with $\mu_\infty$ as characteristic viscosity [21].

## 5.2. $\beta \to \infty$

In the limit $\beta \to \infty$, boundary layer problem is again obtained as in the Newtonian fluid case. Equation (2.5) can be rewritten as

$$8\frac{\partial U_\infty}{\partial T} - \frac{1}{\beta^2}\frac{\partial}{\partial Y}\left\{\left[1 - c + c\left(1 + Cu^2\left(\frac{\partial U_\infty}{\partial Y}\right)^2\right)^{(n-1)/2}\right]\frac{\partial U_\infty}{\partial Y}\right\} - \frac{1}{\beta^2}\cos(T) = 0. \tag{5.5}$$

The solution $U_\infty$ is sought in the two boundary layers, adjacent to the walls $Y = 0$ and $Y = 1$, with thickness $\sim O(\beta^{-1})$ and in the interior region between them as a perturbation expansion in $1/\beta^2$, similarly to (3.8)

$$U_\infty(T, Y) = \frac{U_{\infty 0}}{8\beta^2} + O\left(\frac{1}{\beta^4}\right). \tag{5.6}$$

Since $(\partial U_\infty/\partial Y) \sim O(\beta^{-1})$, instead of the parameter $Cu$ in (2.5), it is better to analyse the solution with respect to the ratio $Cu/\beta$. Then in the boundary layers and in the interior region the solution strongly depends on the value of the ratio Carreau number $Cu$ to Womersley number $\beta$. It occurs that in the limit $Cu/\beta \to 0$, the solution $U_{\infty 0} \to V_{\infty 0}$. For $Cu \gg 1$ and $Cu/\beta \gg 1$, the solution in the boundary layer can be found only numerically or by some approximate methods. At $Cu/\beta \gg 1$, the solution $U_{\infty 0}$ tends numerically to $f_{\infty 0}(T, Y)$, which is the first term in (3.10), i.e.

$$f_{\infty 0}(T, Y) = \sin T - \exp\left(\frac{-2\beta}{\sqrt{1 - c}}Y\right)\sin\left(T - \frac{2\beta}{\sqrt{1 - c}}Y\right)$$
$$- \exp\left(\frac{-2\beta}{\sqrt{1 - c}}(1 - Y)\right)\sin\left(T - \frac{2\beta}{\sqrt{1 - c}}(1 - Y)\right). \tag{5.7}$$

In this case, the boundary layers are thinner (with width $O(\frac{\sqrt{1-c}}{\beta})$) than those of the Newtonian flow.

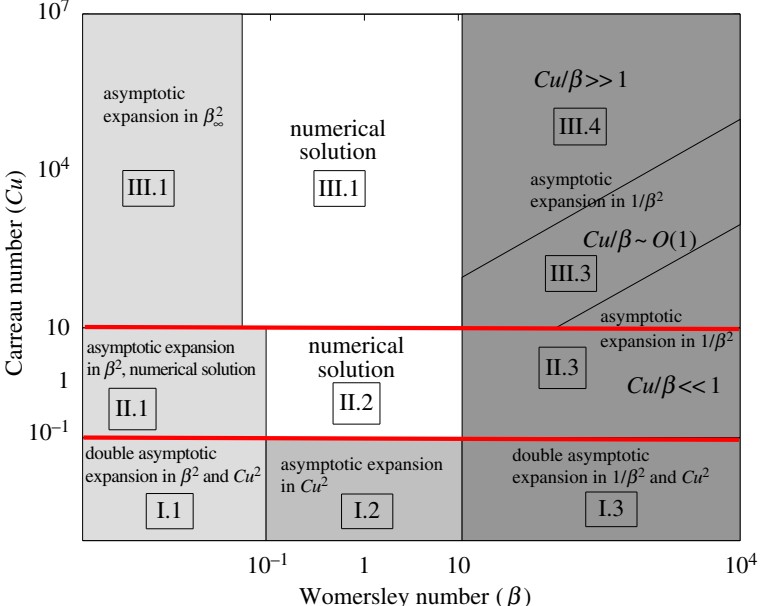

**Figure 3.** Different regimes for the velocity solution $U(T, Y)$ with respect to Womersley number $\beta$ and Carreau number $Cu$: (I) $Cu \ll 1$ (low shear viscosity region)—asymptotic expansion in $Cu^2$ [15]: (I.1) $\beta \ll 1$: $U_0 = U_{00} + O(\beta^2) + O(Cu^2)$, where $U_{00} \to V_{00}$; (I.2) $\beta \sim O(1)$: $U = V + O(Cu^2)$; (I.3) $\beta \gg 1$: $U_\infty = U_{\infty 0}/8\beta^2 + O(1/\beta^4) + O(Cu^2)$, where $U_{\infty 0} \to V_{\infty 0}$; (II) $Cu \sim O(1)$ (transitional shear viscosity region): (II.1) $\beta \ll 1$: $U_0 = U_{00} + O(\beta^2)$, where $U_{00}$—numerical solution; (II.2) $\beta \sim O(1)$—numerical solution; (II.3) $\beta \gg 1$ and $Cu/\beta \ll 1$: $U_\infty = U_{\infty 0}/8\beta^2 + O(1/\beta^4)$, where $U_{\infty 0} \to V_{\infty 0}$; (III) $Cu \gg 1$ (high shear viscosity region): (III.1) $\beta_\infty \ll 1$: $U_0 = U_{00} + O(\beta^2)$, where $U_{00} \to f_{00}/(1-c)$; (III.2) $\beta_\infty \sim O(1)$—numerical solution; (III.3) $\beta \gg 1$ and $Cu/\beta \sim O(1)$: $U_\infty = U_{\infty 0}/8\beta^2 + O(1/\beta^4)$, where $U_{\infty 0}$—numerical solution; (III.4) $\beta \gg 1$ and $Cu/\beta \gg 1$: $U_\infty = U_{\infty 0}/8\beta^2 + O(1/\beta^4)$, where $U_{\infty 0} \to f_{\infty 0}$. The red lines are as in figure 1.

In figure 3, we present a map of Womersley versus Carreau number space, in log-log scale for the different approximations of the velocity $U(T, Y)$. Figure 3 shows the transitional character of $Cu = O(1)$ (in sense of transition between the two Newtonian solutions corresponding to low and high shear), which concerns the Poiseuille and transition flow regime (the flow regime between the viscous Poiseuille flow and inertia Womersley flow). In the Womersley flow regime, the transition between the two Newtonian solutions is given by the effective Carreau number $Cu/\beta = O(1)$.

# 6. Results

In order to illustrate our results, the problem (2.5) and (2.6) has been solved numerically by the Crank–Nickolson method in finite differences. The time interval has been taken long enough (up to $20\pi$ for lower $Cu$ and up to $30\pi$ for higher $Cu$), in order to eliminate the influence of the initial condition (4.8). In the following analysis of the solution, the interval $T \in [18\pi, 20\pi]$ has been considered for $Cu < 1000$ and $T \in [28\pi, 30\pi]$ for $Cu \gtrsim 1000$.

In figure 4, the Carreau solution $U(T, Y)$ and both Newtonian solutions $V(2\pi, Y)$ and $f(2\pi, Y)/(1-c)$ are plotted for different values of $Cu = 10^k$ ($k = 0, 1, \ldots 5$) at fixed $n = 0.5$, $\beta = 0.884$ and $c = 0.999$, and at time $T = 20\pi$ (for $Cu = 10^k$, $k = 0, 1, 2$) and $T = 30\pi$ (for $Cu = 10^k$, $k = 3, 4, 5$). It is seen that for $Cu = 1$ the solutions $U(20\pi, Y)$ and $V(2\pi, Y)$ are almost equal. (Note that $V(T, Y)$ and $f(T, Y)$ are periodic in T with period $2\pi$.) The value 0.884 of $\beta$ corresponds to the blood flow in common carotid artery of diameter 6.65 mm [31].

For the special case of $\beta \to 0$, the Carreau velocity solution together with the Newtonian solutions $V_{00}$ and $f_{00}/(1-c)$ are shown in figure 5 for $c = 0.9$, $n = 0.5$ and $T = 20\pi$ at $Cu \ll 1$ and $Cu \gg 1$. In these examples $\beta = 0.01$, while $Cu = 0.01$ and $Cu = 10^5$, respectively. Then, $\beta_\infty$ is also small, i.e. $\beta_\infty = \beta/\sqrt{1-c} = 0.0316 \ll 1$, such that the asymptotic expansion (3.7) holds. It is well seen in figure 5 that the Carreau velocity is very close to $V_{00}$ at $Cu = 0.01$ and to $f_{00}/(1-c)$ at $Cu = 10^5$.

For high values of $\beta$, the solutions $V(T, Y)$ and $f(T, Y)/(1-c)$ are presented in figure 6 for $c = 0.9$, $T = 2\pi$ and $\beta = 100$ close to the left boundary $Y = 0$ (the plot close to $Y = 1$ is mirror image of this one). As

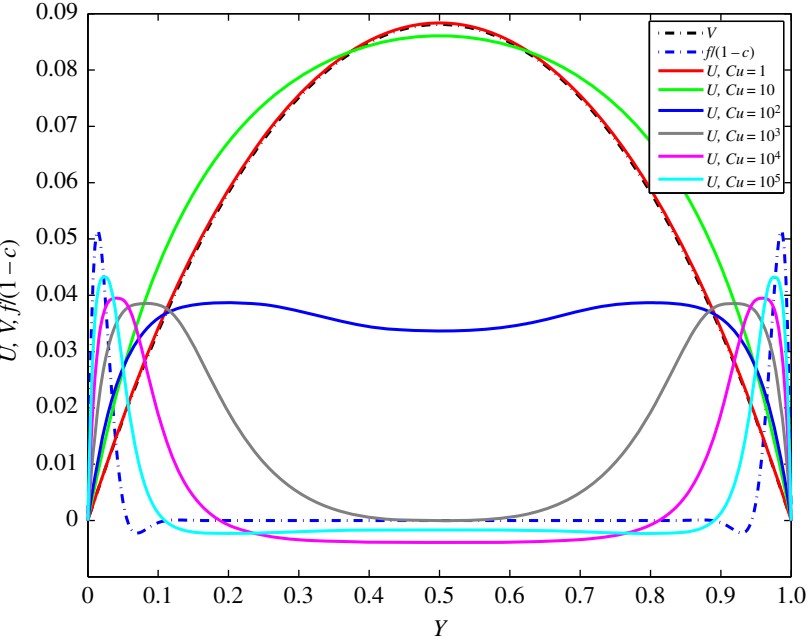

**Figure 4.** Distribution of $U(T, Y)$, $V(2\pi, Y)$ and $f(2\pi, Y)/(1 - c)$ for different values of $Cu = 10^k$ ($k = 0, 1, 2, 3, 4, 5$) at $n = 0.5$, $\beta = 0.884$, $c = 0.999$ and time $T = 30\pi$, as noted in the legend.

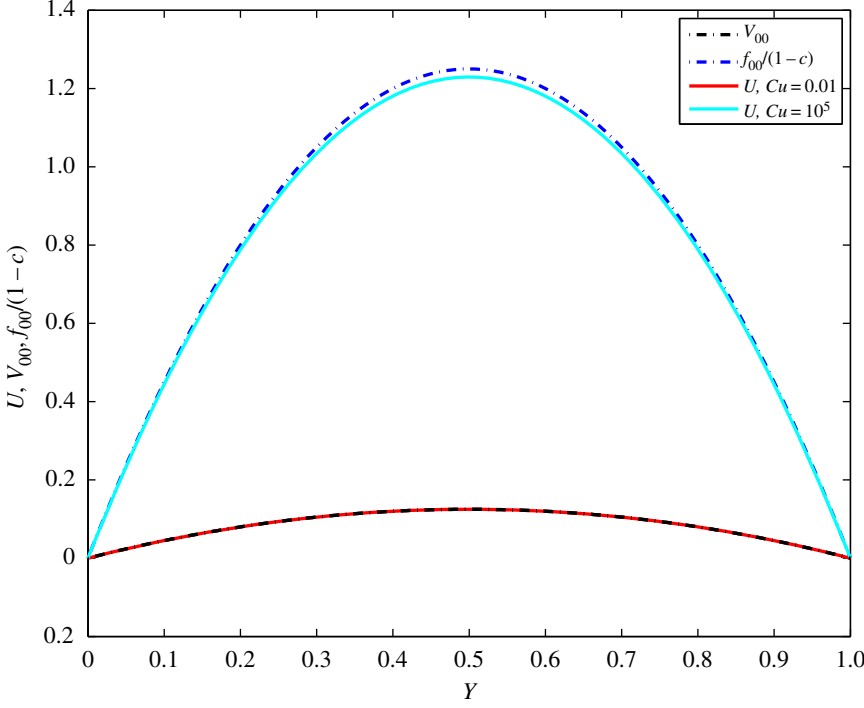

**Figure 5.** Distribution of $V_{00}(2\pi, Y)$, $f_{00}(2\pi, Y)/(1 - c)$ and $U(T, Y)$ at $n = 0.5$, $c = 0.9$, $\beta = 0.01$, $T = 20\pi$ and $Cu = 0.01$, $Cu = 10^5$.

pointed out above, the boundary layer of $f_{\infty0}/(8\beta_\infty^2(1 - c)) = f_{\infty0}/8\beta^2$ is thinner than that of $V_{\infty0}/8\beta^2$. These solutions are used for comparison with the Carreau solution. It occurs that at $Cu \leq 100$ the Carreau solution is very close to $V_\infty$, except in some tiny regions of the boundary layers, which are almost invisible and the plot is not presented here. Analysing the solution $U_\infty(T, Y)$ of (5.5), we arrive to the fact that the function $\Phi$ ($\partial U_\infty/\partial Y$) governs the solution form (if equation (5.5) is rewritten similarly to equation (4.7)), since it takes values between 0 and 1 according to lemma 4.2. If the solution $U_\infty$ is substituted by $U_{\infty0}$ in $\Phi$, then $\Phi(\partial U_{\infty0}/\partial Y)$ differs slightly from 1 for a large range of $Cu/\beta \ll 1$. This means that the solution $U_\infty(T, Y)$ can be approximated by $V_{\infty0}(T, Y)$ up to $O(1/\beta^4)$. Thus, it occurs that, if $Cu/\beta \ll 1$, the solution $U_{\infty0} \approx V_{\infty0}$.

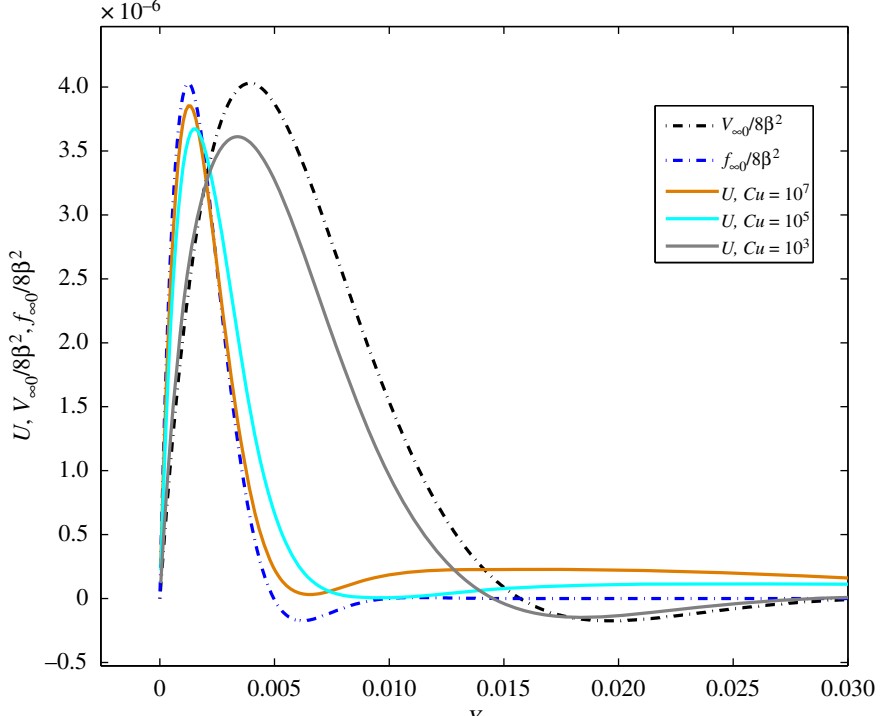

**Figure 6.** Distribution of $U(T, Y)$, $V_{\infty 0}(2\pi, Y)/8\beta^2$ and $f_{\infty 0}(2\pi, Y)/8\beta^2$ at $n = 0.5$, $c = 0.9$, $\beta = 100$ for $Cu = 10^k$ ($k = 3, 5, 7$) at time $T = 30\pi$ in the boundary layer near to $Y = 0$.

In figure 6, the cases of $Cu = 10^k$ ($k = 3, 5, 7$) at $\beta = 100$, i.e. $Cu/\beta > O(1)$, are also considered. From the plots, it can be concluded that with the increase of $Cu$ the solution $U_{\infty 0}$ goes closer to the solution $f_{\infty 0}$ instead of to $V_{\infty 0}$. The calculations have been performed for a longer time interval up to $30\pi$ in order to obtain periodicity.

# 7. Discussion

The proven bounds for the Carreau velocity and its gradients, given by $K_1$ on equations (4.29) and (4.32) and the bounds for the absolute difference between the Newtonian and Carreau velocity solutions—by $K_2$, equations (4.33) and (4.34) are valid for every $\beta \in (0, \infty)$, $n \in (0, 1)$ and $Cu \in (0, \infty)$. However, the bound $K_2$, equations (4.33) and (4.34), is more useful at $Cu \ll 1$ or in the limit $n \to 1$.

Let us analyse the Newtonian velocity solution $f(T, Y)$, given by (17) and (18) of [21] (there denoted by $v$), corresponding to the lower viscosity $\mu_\infty$ as characteristic viscosity, and the solution $U(T, Y)$ of equations (4.7) and (4.8). Since the dimensional velocities are the same, then from equation (27) of [21], the function $U(T, Y)$ satisfies the bound

$$\left| U(T, Y) - \frac{f(T, Y)}{1 - c} \right| \le \frac{c}{8(1 - c)} \quad \text{for } T \ge 0, \quad Y \in [0, 1], \tag{7.1}$$

while the bound given by equation (4.33) with (4.35) concerns the difference between $U(T, Y)$ and $V(T, Y)$. From the two bounds, it follows that $U(T, Y)$ is close to both Newtonian solutions, in the limit $c \to 0$, which is evident since $\mu_\infty \approx \mu_0$, i.e. there exists only one Newtonian velocity solution.

However, in the limit $c \to 1$, but still $c \ne 1$, the bound (7.1) is not appropriate, i.e. does not give any valuable information for the relation between the Carreau and Newtonian velocity solution. In this respect, the bound (4.33) is more suitable as it depends on the other parameters. In figure 7, the bound $K_2$ is plotted for different values of $Cu$ and $n$, at $c = 0.999$. The line $c/2(1 - c)$ is also plotted for comparison with $K_2$, which is an indication that it is not possible to regard the solution difference using only the parameter $c$. It is evident that $K_2$ strongly depends on $n$ and $Cu$ at fixed $c$: decreasing with $n$ and increasing with $Cu$. The behaviour of $K_2$ for other values of $c$ is similar, as increasing with $c$. Here, it must be noted that the bound $K_2$ is only a qualitative measure of the solution difference.

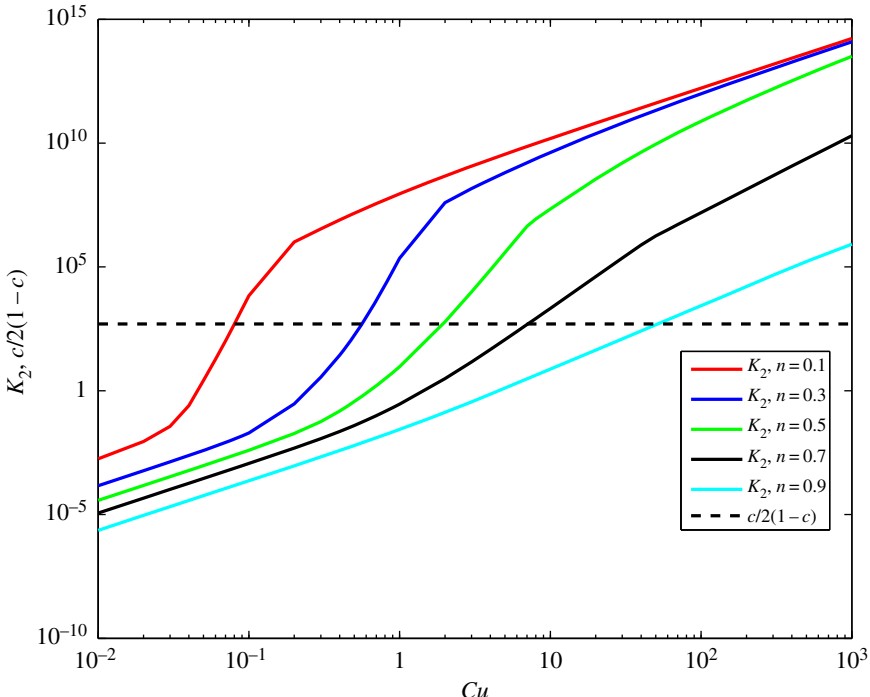

**Figure 7.** The bounds $K_2$ and $c/2(1-c)$ for different $Cu$ at $n$ from 0.1 to 0.9 by 0.2 and $c = 0.999$.

**Table 1.** Maximum differences based on the numerical calculations at $n = 0.5$, $\beta = 0.884$, $Y \in [0, 1]$ and $T \in [18\pi, 20\pi]$ for $Cu = 10^k$ ($k = 0, 1, 2$) or $T \in [28\pi, 30\pi]$ for $Cu = 10^k$ ($k = 3, 4, 5$) for $c = 0.999$ and $c = 0.9$.

| $c$ | $Cu$ | $\max\lvert U - V\rvert$ | $\max\lvert U_Y - V_Y\rvert_{Y=0}$ | $\max\lvert U - \frac{f}{(1-c)}\rvert$ | $\max\lvert U_Y - \frac{f_Y}{(1-c)}\rvert_{Y=0}$ |
|---|---|---|---|---|---|
| 0.999 | 1 | 0.0017 | 0.0157 | 0.163 | 12.21 |
| | 10 | 0.0647 | 0.4932 | 0.1508 | 11.75 |
| | 100 | 0.1294 | 1.2637 | 0.131 | 11.01 |
| | 1000 | 0.1399 | 2.909 | 0.1033 | 9.366 |
| | 10 000 | 0.1374 | 5.82 | 0.0659 | 6.537 |
| | 100 000 | 0.1545 | 8.931 | 0.0324 | 3.407 |
| 0.9 | 1 | 0.0016 | 0.014 | 0.1448 | 0.8544 |
| | 10 | 0.0516 | 0.358 | 0.1168 | 0.5131 |
| | 100 | 0.1155 | 0.6463 | 0.0525 | 0.2327 |
| | 1000 | 0.137 | 0.7645 | 0.0195 | 0.1223 |
| | 10 000 | 0.1431 | 0.8275 | 0.0066 | 0.0449 |
| | 100 000 | 0.1447 | 0.8534 | 0.0021 | 0.0149 |

Furthermore, we shall give some quantitative bounds coming from the numerical calculations, with which the established tendencies of the bounds (4.33), (4.34) and (4.35) will be confirmed.

In table 1, the maximal differences between $U(T, Y)$ and $V(T, Y)$ and between their boundary gradients in the corresponding intervals, are presented for $n = 0.5$ and $\beta = 0.884$ at $c = 0.999$ (for all the cases of figure 4) and $c = 0.9$. In fact, at $Cu = 0$ the calculations show that $\max_{T \in [18\pi, 20\pi], Y \in [0,1]} \lvert U(T, Y) - V(T, Y)\rvert \leq 0.00174$, as seen in table 1. With the increase of $Cu$, the Carreau solution begins to deviate from $V(T, Y)$ and approaches $f(T, Y)/(1-c)$. This behaviour is observed also for other values of $c$, $n$ and times $T$. It is interesting to note, that at $n \to 0$ the Carreau solution is much closer to $f(T, Y)/(1-c)$, while at $n \to 1$ it becomes exactly the Newtonian solution $V(T, Y)$, which is evident from equation (2.5). It is clear that the Carreau number is responsible for the change of solution behaviour for both values of $c$. As predicted by the bound $K_2$ (4.35), the Carreau solution $U(T, Y)$ becomes closer to the Newtonian one $V(T, Y)$ with the increase of $n$ and decrease of $c$ and $Cu$. The gradient differences on

the channel wall ($Y = 0$ or $Y = 1$) are very important when analysing the wall shear stresses (WSS), and deciding with which Newtonian solution the flow can be eventually approximated. For example, at $c = 0.9$ the solution $U(T, Y)$ can be approximated with $f(T, Y)/(1 - c)$, when their gradient difference is small enough, e.g. equal to 0.0149 in the case of $Cu = 10^5$.

Finally, we could make the following statement, that the effective Carreau number is $Cu$ in the Poiseuille and transition regime and $Cu/\beta$ in the Womersley flow regime. This leads to the conjecture: basically, the effective Carreau number is responsible for solution type changes, converging to one or the other Newtonian solution. The other parameters: Womersley number $\beta$, $n$ and $c$ can only accelerate or delay this convergence process when increasing the effective Carreau number.

In order to support this statement, we reconsider the example cases of our previous work [21] for flows in a channel with width 5 mm. The case of blood, shown in fig. 2a of [21], corresponds to $Cu = 1775$, $c = 0.938$, $n = 0.357$ and $\beta = 0.649$. The Carreau velocity profile is closer to the Newtonian velocity at viscosity $\mu_\infty$, i.e. to $f(T, Y)/(1 - c)$. However, the presented case of the polymer solution HEC 0.5% in fig. 2b of [21], corresponding to $Cu = 9$, $c = 0.995$, $n = 0.5088$ and $\beta = 0.327$, is closer to the Newtonian velocity with the higher viscosity $\mu_0$, i.e. to $V(T, Y)$. These examples show that the big difference in the Carreau number in both cases leads to a big difference of their velocity profiles. The corresponding $\text{WSS} = \mu_c B/H |\partial U/\partial Y|_{Y=0,Y=1}$, which are very important for practical applications, also show the same tendency (fig. 3a, b in [21]). In the blood case, the obtained peak WSS corresponding to that in a human brachial artery (with diameter 5 mm) is 4.25 Pa, which is close to the experimental limits [31,32]: $3.3 \pm 0.7$ Pa for an artery of diameter $4.4 \pm 0.6$ mm. The obtained peak WSS of the Carreau model is slightly higher than the WSS of the Newtonian model calculated with the lower viscosity $\mu_\infty$, which is 4.04 Pa. It is worthwhile to mention that the peak WSS of the Newtonian model, calculated with the higher viscosity $\mu_0$, is 14.24 Pa, which is far away from the experimental data. This result is very important to support our conjecture that the fluid can be approximated as a Newtonian fluid with the lower viscosity, if the Carreau number is high enough. In the cases when this viscosity is unknown (hardly to be measured), the flow solution remains non-Newtonian, described by the Carreau model, as done in the present work, or by another appropriate model.

From the obtained results, we can conclude that the flow remains laminar, i.e. the peak Reynolds numbers defined as $Re_{\max} = \rho \bar{V} H / \min(\mu_c) \ll Re_{cr}$, where $\bar{V}$ is the maximal mean cross-sectional velocity, corresponding to maximal volume flow rate in time. The experimental observations of Patel & Head [33] show that the approximate value of 1300 may be accepted as the lower critical Reynolds number for steady channel flow. We suspect that for oscillatory channel flow the critical Reynolds number will depend on the Womersley number, as it has been reported for Newtonian flows in tubes [34], and to be higher or around the critical Reynolds number for steady flows. In the two examples of blood and HEC solution flows, as cited above, the obtained peak $Re_{\max}$ are 850 and 14.2, respectively.

We expect that the critical Reynolds number, $Re_{cr}$, of Carreau fluid flows in straight channels will not be very different from 1300. Although that we have not found any experimental confirmation for $Re_{cr}$ of shear-thinning flows in channels, there are many results for circular pipes, that support this idea. For example, the experiments show that the transition to turbulence of shear-thinning flows in pipes may be delayed in comparison to Newtonian fluids [35].

## 8. Conclusion

The oscillatory flow of a Carreau fluid in a straight infinite channel has been studied in comparison to the two limiting cases of Newtonian fluids (with higher or lower viscosity). The longitudinal velocity is a solution of a parabolic nonlinear equation, which depends on the Carreau and Womersley numbers. An analysis of the non-linearity of the Carreau problem has been performed with respect to the Womersley number: low, high and intermediate. For the first two cases, asymptotic expansions are proposed for the Carreau flow velocity. Since for the intermediate Womersley numbers the Carreau flow velocity cannot be found in an analytic form, theoretical bounds have been proven with theorems depending on the other parameters of the Carreau viscosity model: Carreau number, Womersley number, power coefficient and the Newtonian viscosity ratio. The theoretical bounds also concern the differences between the Newtonian and Carreau flow velocity and between their gradients on the channel wall but have only a qualitative character. To give some quantitative results for these differences, the Carreau flow velocity problem has been solved numerically. Its solution shows that these differences increase with the Carreau number, for any value of the Womersley number (Reynolds number being in the limits of laminar flow). Therefore, we can state the following

conjecture: that the effective Carreau number only is responsible for the different behaviour of the velocity solution. For practical purposes, it is very important which one of the Newtonian velocities to be used in the case to simplify the problem, i.e. if it is possible for the Carreau velocity to be approximated with one of the limiting Newtonian velocities, corresponding to the lower and higher viscosity. For example, in the case of a blood flow, which has been studied in our previous works, the Carreau number is high enough for the solution to be approximated by the Newtonian solution corresponding to the lower viscosity $\mu_\infty$, as their peak velocities are very close, and the WSS are in the limits of the experimentally measured ones for the brachial human artery [31,32].

The obtained estimates can serve as an indicator to what extent the considered problem may have one or another asymptotic solution corresponding to a developed flow in a channel. As the asymptotic solutions are given by simple expressions, they can be easily implemented as initial or boundary velocity profiles when solving more complicated problems in complex geometries by professional or home-made software.

The flows in perfectly straight two-dimensional channels are considered in the present work. In fact, small disturbances of the channel width can be added to the model. Then the dimensionless wall position will be given as $Y = -\varepsilon g(T)$ and $Y = 1 + \varepsilon g(T)$, where $|g(T)| \leq 1$ and $\varepsilon \ll 1$. In this way, the present problem will occur in the zero-th order approximation in $\varepsilon$ of the more general problem of wall perturbations in time (from elastic or other sources). For example, the proper knowledge of the flow velocity in rigid channels/tubes is a starting point before the introduction of elasticity in the model, as a fluid-structure interaction.

Another further continuation of the present work is to use a more general function $G(T)$ of the pressure gradient instead of the pure oscillation in equation (2.5). $G(T)$ must be bounded and smooth enough. The obtained bounds for the Carreau velocity will be similar, but it is necessary to know explicitly the function $G(T)$. Moreover, for general function $G(T)$, the solution of the Newtonian velocity cannot be given in a closed analytic form like in the present work for $G(T) = \cos T$.

Finally, we point out the open problem connected with the special case of $c = 1$. Then the equation (2.5) is not a uniformly parabolic one and the existence of a classical solution is questionable. In this case it is possible for the gradient $\partial U/\partial Y$ on the boundaries $Y = 0$ and $Y = 1$ to become infinite for some times $T$. However, for $c \in [0, 1)$, the gradient $\partial U/\partial Y$ cannot reach infinite values anywhere inside the region $Y \in [0, 1]$ and $T \geq 0$ according to corollary 4.5. Our conjecture is that at $c = 1$ and $Cu \to 0$, the gradient is bounded and the treated problem still has a classical solution.

Data accessibility. This work does not have any additional data.
Authors' contributions. All authors contributed equally to all aspects of the research and gave their final approval for publication.
Competing interests. The authors declare no competing interests.
Funding. The author N.K. has been partially supported by the grant no. BG05M2OP001-1.001-0003, financed by the Science and Education for Smart Growth Operational Program (2014–2020).
Acknowledgements. We are grateful to the anonymous reviewers for their comments to improve the manuscript.

# Appendix A

*Proof.* From the inequalities $\Phi(\eta) \geq n(1 + Cu^2\eta^2)^{(n-1)/2}$ and $1/2(1 - c) \geq 1$ for $1 > c \geq 0.5$, we get the following sequence of inequalities:

$$K_1^1 \leq \frac{1}{2}\left[1 - c + cn\left(1 + \frac{Cu^2}{4(1 - c)^2}\right)^{(n-1)/2}\right]^{-1}$$

$$\leq \frac{1}{2}\left(1 + \frac{Cu^2}{4(1 - c)^2}\right)^{(1-n)/2}\left[(1 - c)\left(1 + \frac{Cu^2}{4(1 - c)^2}\right)^{(1-n)/2} + cn\right]^{-1}$$

$$\leq \frac{1}{2n}\left(1 + \frac{Cu^2}{4(1 - c)^2}\right)^{(1-n)/2} \leq \frac{1}{2n}\left[\frac{1}{4(1 - c)^2} + \frac{Cu^2}{4(1 - c)^2}\right]^{(1-n)/2}$$

$$\leq \frac{1}{2n}(1 + Cu^2)^{(1-n)/2}\left(\frac{1}{2(1 - n)}\right)^{1-n}\left(1 + \frac{Cu^2}{4n^2}\right)^{1-n} \tag{A 1}$$

which proves the bound (4.21). □

# Appendix B

**Proof.** Simple computations give us for $c \geq 0.5$ and from (A 1) the sequence of inequalities

$$K_1^{m+1} = \frac{1}{2}[1 - c + c\Phi(K_1^m)]^{-1} \leq \frac{1}{2}\{1 - c + cn[1 + Cu^2(K_1^m)^2]^{(n-1)/2}\}^{-1}$$

$$\leq \frac{1}{2n}\{1 - c + c[1 + Cu^2(K_2^m)^2]^{(n-1)/2}\}^{-1}$$

$$= \frac{1}{2n}[1 + Cu^2(K_1^m)^2]^{(1-n)/2}\{c + (1-c)[1 + Cu^2(K_1^m)^2]^{(1-n)/2}\}^{-1} \leq \frac{1}{2n}[1 + Cu^2(K_1^m)^2]^{(1-n)/2}$$

$$\leq \frac{1}{2n}\left[1 + \frac{Cu^2}{4n^2}\left(\frac{1}{2(1-c)}\right)^{2(1-n)^m}(1 + Cu^2)^{(1-n)^m}\left(1 + \frac{Cu^2}{n^2}\right)^{2\sum_{j=1}^{m}(1-n)^j}\right]^{\frac{1-n}{2}}$$

$$\leq \frac{1}{2n}\left(\frac{1}{2(1-c)}\right)^{(1-n)^{m+1}}(1 + Cu^2)^{\frac{1}{2}(1-n)^{m+1}}\left(1 + \frac{Cu^2}{4n^2}\right)^{\sum_{j=1}^{m+1}(1-n)^j}$$

and (4.26) is proved for $m + 1$. □

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
