## [Reviewer comments · Royal Society Open Science]

Review History

RSOS-191305.R0 (Original submission)

Review form: Reviewer 1 (Richard Clarke)

Is the manuscript scientifically sound in its present form?

Yes

Are the interpretations and conclusions justified by the results?

Yes

Do you have any ethical concerns with this paper?

No

Have you any concerns about statistical analyses in this paper?

No

Recommendation?

Major revision is needed (please make suggestions in comments)

Comments to the Author(s)

Please see the attached (Appendix A).

Review form: Reviewer 2

Is the manuscript scientifically sound in its present form?

No

Are the interpretations and conclusions justified by the results?

Yes

Do you have any ethical concerns with this paper?

No

Have you any concerns about statistical analyses in this paper?

No

Recommendation?

Major revision is needed (please make suggestions in comments)

Comments to the Author(s)

See attached file (Appendix B).

Decision letter (RSOS-191305.R0)

02-Oct-2019

Dear Dr Tabakova,

The editors assigned to your paper ("Oscillatory Carreau flows in straight channels") have now received comments from reviewers. We would like you to revise your paper in accordance with the referee and Associate Editor suggestions which can be found below (not including confidential reports to the Editor). Please note this decision does not guarantee eventual acceptance.

Please submit a copy of your revised paper before 25-Oct-2019. Please note that the revision deadline will expire at 00.00am on this date. If we do not hear from you within this time then it will be assumed that the paper has been withdrawn. In exceptional circumstances, extensions may be possible if agreed with the Editorial Office in advance. We do not allow multiple rounds of revision so we urge you to make every effort to fully address all of the comments at this stage. If deemed necessary by the Editors, your manuscript will be sent back to one or more of the original reviewers for assessment. If the original reviewers are not available, we may invite new reviewers.

- Data accessibility

<http://datadryad.org/submit?journalID=RSOS&manu=RSOS-191305>

- Competing interests

- Authors' contributions

- Acknowledgements

- Funding statement

Best regards,
Lianne Parkhouse
Royal Society Open Science
openscience@royalsociety.org

on behalf of Dr Oliver Jensen (Associate Editor) and Mark Chaplain (Subject Editor)
openscience@royalsociety.org

Editorial Comments:

A number of language editing services are available for authors whose first language is not English. We recommend that you ask a native speaker of English or solicit the support of a language editing service (<https://royalsociety.org/journals/authors/language-polishing/>) prior to resubmitting the manuscript. If requested to edit the written English, you must provide proof that you have done so: acceptable proof includes a certificate of language-editing from a language editing service or a signed letter from a native speaker of English. If you do not provide this proof, your manuscript may be returned to you upon resubmission.

Associate Editor's comments (Dr Oliver Jensen):

The reviewers make a number of constructive suggestions for improving and clarifying your manuscript. Please consider carefully all the points raised and revise your manuscript accordingly.

Reviewers' Comments to Author:

Reviewer: 1
Comments to the Author(s)

Please see the attached.

Reviewer: 2
Comments to the Author(s)

See attached file.

Author's Response to Decision Letter for (RSOS-191305.R0)

See Appendix C.

RSOS-191305.R1 (Revision)

Review form: Reviewer 1 (Richard Clarke)

Is the manuscript scientifically sound in its present form?

Yes

Are the interpretations and conclusions justified by the results?

Yes

Is the language acceptable?

Yes

Do you have any ethical concerns with this paper?

No

Have you any concerns about statistical analyses in this paper?

No

Recommendation?

Accept with minor revision (please list in comments)

Comments to the Author(s)

I thank the authors for their responses and clarifications in the manuscript.

I still feel as though the authors have somewhat missed an opportunity in not applying their analysis to non-oscillatory flows, even if it requires a partial numerical treatment. It would certainly help to move the applicability of the analysis closely to the stated area of interest, i.e. blood flow, which of course is not perfectly oscillatory.

I note the authors' response as to whether or not the conclusions about the Carreau number controlling the transition are obvious, but feel as though their arguments perhaps need fleshing out a little more. In particular, providing greater insights into situations when dU/dY would tend to zero in the presence of no-slip boundaries. I also would have thought that a more interesting case would be when Cu tends to zero but dU/dY tends to infinity. However, I am guessing that for the straight channel problem neither of these cases are likely, and so in this context it still seems as though the conclusions are somewhat obvious for the problem being solved. I will not labour this point further, but the authors may like to reflect more about these more interesting scenarios in their Conclusion sections.

With regards the value of beta, and would still encourage them to reflect on whether 4 decimal places is entirely appropriate given the idealisations already in their model.

Overall, in my view the very simple problem considered here probably to not best bring out the value of the analysis, but I have no strong objections to publication.

Review form: Reviewer 3

Is the manuscript scientifically sound in its present form?

No

Are the interpretations and conclusions justified by the results?

No

Is the language acceptable?

Yes

Do you have any ethical concerns with this paper?

No

Have you any concerns about statistical analyses in this paper?

No

Recommendation?

Reject

Comments to the Author(s)

My appreciation is that this is a rather mathematical paper with little physical content. I don't know whether that is appropriate for this journal, Royal Soc. Open Science??

The authors derive bounds for velocity gradients and velocity itself, by approximation to solutions of an oscillatory flow in a planar channel due to an imposed sinusoidal pressure gradient, for a fluid whose viscosity follows the Carreau model. For that, they expand the (unknown) solution in terms of Beta (the so-called Womersley number, which is the square-root of the dimensionless frequency), when Beta is small, or $1/\text{Beta}$ when Beta is large. This, for me, is a strange procedure because the basic steady solution for Poiseuille flow (for $\text{Beta}=0$) is unknown for the Carreau model. So in the perturbation method utilized, when the governing equation (or the "expected" solution) is expanded in a series in terms of the small parameter Beta, the first element of the series is not known. The corresponding procedure for the Newtonian case is well illustrated in White's book (see below), showing the occurrence of the Richardson's annular effect (similar to the velocity profiles given here in Figs 4 and 6). I am surprised to notice that even these classical references are not cited.

One is tempted to ask the authors why haven't they tried the same problem with, for example, the power-law viscosity model? At least they would know the solution for $\text{Beta}=0$. (I have not check the literature to see if that problem has already been considered by other authors).

In addition, I think the authors have not done an adequate literature review. The solution for the Newtonian fluid is a classical solution in fluid mechanics (or mathematical physics), given for example in the book of White (Viscous Fluid Flow). However the authors present it (Eqs. 3.1-3.3) as if it was derived by themselves. Many other authors have used that solution, for example Duarte et al (J Non-Newt Fluid Mech. 154 (2008) 153-), who also considered the pulsating flow of viscoelastic fluids, and in particular with the Carreau model, Miranda et al (Int J Num Meth Fluids 57 (2008) 295). As noted above, the classical papers, such as Sx1 (1930) (Z Phy 61, 349-, first solution for pipe flow), Richardson & Tyler (1929) (Proc Phy Soc London, 42, p. 1-15, found the velocity overshoot near wall at large frequencies), are not given.,

This problem has too many dimensionless numbers (the Womersley number, the Carreau number Cu - which has little physical meaning, a viscosity ratio here called c , the power law index n) and so in a future study I suggest the authors concentrate in typical values of these parameter, eg. $N=0.5$, $c=0.99$ or 0.9 , Cu of order 10, 100, and seek the effect of the imposed frequency. It should be aid that obtaining a numerical solution for this problem is straightforward and so the expected trends at high/low Beta, or Cu , are easy to find.

The values of the Carreau number here employed are unrealistic (for example in Table 1 and Figs. 4,5,6). The authors have failed to recognize that Cu is not independent of n and c . If $Cu = \lambda \cdot \dot{\gamma}$ ($\lambda =$ time constant of Carreau model; $\dot{\gamma} =$ typical shear

rate of the flow) is smaller than 1 or greater than a certain limit $Cu_{\infty} = \lambda \cdot \gamma_{\text{max}}$, they simply have a Newtonian fluid with viscosity η_0 or η_{∞} , respectively. This explains their findings at high and low β 's and Cu 's. From simple calculations I get $\log(Cu_{\infty}) = \log(1-c)/(n-1)$. For $c=0.9$, $n=0.5$ this gives $Cu_{\infty} = 100$. Using Cu greater than this value gives results for a Newtonian fluid with viscosity η_{∞} .

Minor points:

- Page 3, calling λ the relaxation time is misleading; λ is just a constant of the model with units of time. $1/\lambda$ is the shear rate at which the viscosity starts to decrease.
- Page 4, Eqs 2.2 and 2.3 are not necessary; suffices to give the momentum equation that is actually solved (a much simpler version of the Navier-Stokes equations)
- Page 4, Eq. 2.4 is wrong (this is probably just a mistype)
- Fig. 4, why the need to specify the time $T=20\pi$? The flow should be sinusoidal (stationary in time) and the exact time moment is irrelevant (except inside a cycle)

Decision letter (RSOS-191305.R1)

21-Feb-2020

Dear Dr Tabakova:

On behalf of the Editors, I am pleased to inform you that your Manuscript RSOS-191305.R1 entitled "Oscillatory Carreau flows in straight channels" has been accepted for publication in Royal Society Open Science subject to minor revision in accordance with the referee suggestions. Please find the referees' comments at the end of this email.

The reviewers and Subject Editor have recommended publication, but also suggest some minor revisions to your manuscript. Therefore, I invite you to respond to the comments and revise your manuscript.

- Ethics statement

- Data accessibility

<http://datadryad.org/submit?journalID=RSOS&manu=RSOS-191305.R1>

- Competing interests

- Authors' contributions

- Acknowledgements

- Funding statement

Because the schedule for publication is very tight, it is a condition of publication that you submit the revised version of your manuscript before 01-Mar-2020. Please note that the revision deadline will expire at 00.00am on this date. If you do not think you will be able to meet this date please let me know immediately.

- 1) A text file of the manuscript (tex, txt, rtf, docx or doc), references, tables (including captions) and figure captions. Do not upload a PDF as your "Main Document".
- 2) A separate electronic file of each figure (EPS or print-quality PDF preferred (either format should be produced directly from original creation package), or original software format)

- 3) Included a 100 word media summary of your paper when requested at submission. Please ensure you have entered correct contact details (email, institution and telephone) in your user account
- 4) Included the raw data to support the claims made in your paper. You can either include your data as electronic supplementary material or upload to a repository and include the relevant doi within your manuscript
- 5) All supplementary materials accompanying an accepted article will be treated as in their final form. Note that the Royal Society will neither edit nor typeset supplementary material and it will be hosted as provided. Please ensure that the supplementary material includes the paper details where possible (authors, article title, journal name).

on behalf of Dr Oliver Jensen (Associate Editor) and Mark Chaplain (Subject Editor)
openscience@royalsociety.org

Associate Editor Comments to Author (Dr Oliver Jensen):

Please make further minor revisions to your paper, addressing the points raised by both reviewers. In particular, please make sure you cite relevant prior literature cited by referee 2.

Reviewer comments to Author:

Reviewer: 1

Comments to the Author(s)

I thank the authors for their responses and clarifications in the manuscript.

I still feel as though the authors have somewhat missed an opportunity in not applying their analysis to non-oscillatory flows, even if it requires a partial numerical treatment. It would certainly help to move the applicability of the analysis closely to the stated area of interest, i.e. blood flow, which of course is not perfectly oscillatory.

I note the authors' response as to whether or not the conclusions about the Carreau number controlling the transition are obvious, but feel as though their arguments perhaps need fleshing out a little more. In particular, providing greater insights into situations when dU/dY would tend to zero in the presence of no-slip boundaries. I also would have thought that a more interesting case would be when Cu tends to zero but dU/dY tends to infinity. However, I am guessing that for the straight channel problem neither of these cases are likely, and so in this context it still seems as though the conclusions are somewhat obvious for the problem being

solved. I will not labour this point further, but the authors may like to reflect more about these more interesting scenarios in their Conclusion sections.

With regards the value of beta, and would still encourage them to reflect on whether 4 decimal places is entirely appropriate given the idealisations already in their model.

Overall, in my view the very simple problem considered here probably to not best bring out the value of the analysis, but I have no strong objections to publication.

Reviewer: 3

Comments to the Author(s)

My appreciation is that this is a rather mathematical paper with little physical content. I don't know whether that is appropriate for this journal, Royal Soc. Open Science??

The authors derive bounds for velocity gradients and velocity itself, by approximation to solutions of an oscillatory flow in a planar channel due to an imposed sinusoidal pressure gradient, for a fluid whose viscosity follows the Carreau model. For that, they expand the (unknown) solution in terms of Beta (the so-called Womersley number, which is the square-root of the dimensionless frequency), when Beta is small, or $1/\text{Beta}$ when Beta is large. This, for me, is a strange procedure because the basic steady solution for Poiseuille flow (for $\text{Beta}=0$) is unknown for the Carreau model. So in the perturbation method utilized, when the governing equation (or the "expected" solution) is expanded in a series in terms of the small parameter Beta, the first element of the series is not known. The corresponding procedure for the Newtonian case is well illustrated in White's book (see below), showing the occurrence of the Richardson's annular effect (similar to the velocity profiles given here in Figs 4 and 6). I am surprised to notice that even these classical references are not cited.

One is tempted to ask the authors why haven't they tried the same problem with, for example, the power-law viscosity model? At least they would know the solution for $\text{Beta}=0$. (I have not check the literature to see if that problem has already been considered by other authors).

In addition, I think the authors have not done an adequate literature review. The solution for the Newtonian fluid is a classical solution in fluid mechanics (or mathematical physics), given for example in the book of White (*Viscous Fluid Flow*). However the authors present it (Eqs. 3.1-3.3) as if it was derived by themselves. Many other authors have used that solution, for example Duarte et al (*J Non-Newt Fluid Mech.* 154 (2008) 153-), who also considered the pulsating flow of viscoelastic fluids, and in particular with the Carreau model, Miranda et al (*Int J Num Meth Fluids* 57 (2008) 295). As noted above, the classical papers, such as Sexl (1930) (*Z Phy* 61, 349-, first solution for pipe flow), Richardson & Tyler (1929) (*Proc Phy Soc London*, 42, p. 1-15, found the velocity overshoot near wall at large frequencies), are not given.,

This problem has too many dimensionless numbers (the Womersley number, the Carreau number Cu - which has little physical meaning, a viscosity ratio here called c , the power law index n) and so in a future study I suggest the authors concentrate in typical values of these parameter, eg. $N=0.5$, $c=0.99$ or 0.9 , Cu of order 10, 100, and seek the effect of the imposed frequency. It should be aid that obtaining a numerical solution for this problem is straightforward and so the expected trends at high/low Beta, or Cu , are easy to find.

The values of the Carreau number here employed are unrealistic (for example in Table 1 and Figs. 4,5,6). The authors have failed to recognize that Cu is not independent of n and c . If $Cu = \lambda \cdot \dot{\gamma}$ (λ = time constant of Carreau model; $\dot{\gamma}$ = typical shear rate of the flow) is smaller than 1 or greater than a certain limit $Cu_{\infty} = \lambda \cdot \dot{\gamma}_{\infty}$, they simply have a Newtonian fluid with viscosity η_0 or η_{∞} , respectively. This explains their findings at high and low Beta's and Cu 's.

From simple calculations I get $\log(Cu_{inf}) = \log(1-c)/(n-1)$. For $c=0.9$, $n=0.5$ this gives $Cu_{inf} = 100$. Using Cu greater than this value gives results for a Newtonian fluid with viscosity $= \eta_{inf}$.

Minor points:

- Page 3, calling λ the relaxation time is misleading; λ is just a constant of the model with units of time. $1/\lambda$ is the shear rate at which the viscosity starts to decrease.
- Page 4, Eqs 2.2 and 2.3 are not necessary; suffices to give the momentum equation that is actually solved (a much simpler version of the Navier-Stokes equations)
- Page 4, Eq. 2.4 is wrong (this is probably just a mistype)
- Fig. 4, why the need to specify the time $T=20\pi$? The flow should be sinusoidal (stationary in time) and the exact time moment is irrelevant (except inside a cycle)

Author's Response to Decision Letter for (RSOS-191305.R1)

See Appendix D.

Decision letter (RSOS-191305.R2)

18-Mar-2020

Dear Dr Tabakova,

It is a pleasure to accept your manuscript entitled "Oscillatory Carreau flows in straight channels" in its current form for publication in Royal Society Open Science. The comments of the reviewer(s) who reviewed your manuscript are included at the foot of this letter.

on behalf of Dr Oliver Jensen (Associate Editor) and Mark Chaplain (Subject Editor)
openscience@royalsociety.org

Appendix A

Oscillatory Carreau flow in straight channels by Tabakova et al. studies the flow of both a Newtonian and non-Newtonian fluid through a straight 2D channel, examining certain limiting cases where asymptotic approximations are appropriate, as well as undertaking analysis to place bounds on the flow as a function of the important non-dimensional parameters.

The analysis presented in Sections 3 and 4a appears to follow quite closely analysis in Ref [14], where asymptotic forms of the flow solution are compared to full numerical solutions.

However, the bounds on the Carreau flow and its gradients presented in Section 4b appears to be wholly new, and I found some interest in the bounds given by Theorems 4.3 and 4.4, which detail the dependence upon both viscosity ratio, as well as the Carreau number.

I believe that the ability to place bounds on such flows is potentially useful, although my principal concern is the very limited regime under which these results apply, i.e. perfectly straight 2D channels. I appreciate the analysis here seems to be highly reliant on a unidirectional flow, but I wonder whether considering the influence of asymptotically-small wall perturbations might expand the study's applicability while still leaving it accessible to the kind of analysis being applied? In its current limited form, I just wonder whether it sufficiently advances our understanding of these kinds of flows through channels.

This is perhaps also compounded by the final conjecture concerning the Carreau number being solely responsible for the transition of the flow between the two Newtonian regimes. Unless I am missing something, is this not immediately obvious from the non-dimensional form of (2.1)?

I do have some other remarks and/or questions that the authors may like to consider:

1. The authors appear to consider a general time dependence, but when comparing between theory and numerical simulations allow transients to decay leaving only purely oscillatory motion. However, there are plenty of applications where transients are of interest (including in blood flow), and so if the bounds are valid both for oscillatory and non-oscillatory transient flows this could be worth exploring more.
2. On page 4, line 8: Should $1/n$ read $1/\omega$, given the definition of beta below?
3. There were some statements on p14, the logic of which I was not able to completely follow. For example, why does $\Phi(dU_\infty/dy)$ taking values between 0 and 1 combined with Lemma 4.2 mean that it governs the solution? Also, why does $\Phi(dV_\infty/dy)$ being differently slightly from 1 means that U_∞ can be approximated by V_∞ up to the stated order of accuracy? Some clarification on these statements would be useful.
4. Why in Figure 2 is this very particular value of beta considered, it seems curious that the number has been specified up to 4 decimal places?
5. It might be interesting to have K_1^0 and K_1^1 plotted on the same axes as a function of c , to gauge the values of c and which these diverge.

6. I am not sure I am convinced by the case for the flow remaining laminar, since the critical Reynolds number cited from Patel and Head does not appear to come from non-Newtonian flow experiments.

Appendix B

Referee's Comments to the Authors

Manuscript Title: Oscillatory Carreau Flows in tright Channels

Author(s): S Tabakova, N Kutev & St Radev

Overview

I found my first read through of this manuscript quite difficult for main two reasons.

First, I did not find it the manuscript's purpose and expected outputs to be entirely clear on my first reading. There did not appear to be a clear narrative running through it, and I wasn't sure in advance exactly what to expect from the different sections. I think some clearer statements in the introduction would help here, along with some re-ordering of the material, and also a better description of the different flow regimes in parameter space. Further details on these points are provided below.

Secondly there were many minor English language errors and non-standard use of mathematical terminology. I accept that this is hard for non-native speakers, but I think this needs to be addressed prior to publication. (Some of the more significant errors are listed below.)

There were also a small number of technical, notational and layout issues that I would like to see the authors address.

Overall, I think the manuscript has the potential to be publishable, but I would like to see significant changes, as discussed below, and a further review before this happens.

Layout, structure and narrative issues

The manuscript would be much improved if it were clearer in the abstract and introduction exactly what it was trying to achieve. I would also suggest that it would be helpful if section 5 was split into two for the two different limits, and if both of these sections came before section 4.

I think figure 1 should be shown and described much sooner, to help readers understand the parameter space. In addition, I would like to see another similar figure, showing the different flow regimes across the same parameter space. The flow regime can be categorised in two ways: First, the flow profile can either be viscous-dominated Poiseuille flow, inertia-dominated Womersley flow, or somewhere between the two. Secondly, the viscosity can either be dominated by μ_0 for low shear, be dominated by μ_∞ for high shear, or be a mixture of the two. Showing the regions occupied by these different regimes on a parameter-space diagram would be most helpful.

In terms of following the work and understanding what is being done and why, I think it would be very helpful to readers if it were explicitly stated (if I

have have understood things correctly) that the objective is to obtain bounds on the velocity and velocity gradient; either on their absolute magnitude, or on the deviation from a comparable Newtonian solution. Setting this out more clearly at the beginning would have helped me greatly when reading the heavy-going section 4 of the manuscript. It would also be good if the authors could say a bit more about why these bounds are useful / where they might be used, and also how tight they are. Could a summary be provided of which of the bounds are most useful in which regions of parameter space?

Technical and scientific issues

- I do not think it is completely obvious that the $Cu \gg 1$ solutions necessarily tend to the Newtonian solution with viscosity μ_∞ (e.g. page 5, line 41). Is it not possible that the small regions where the shear rate is small have a global effect on the flow profile? (It should be noted that the spatial extent of these regions is time-dependent, and for short periods of time in each cycle the spatial regions will not be small.) I think more should be said here, perhaps with an order of magnitude estimate (in terms of Cu) for the errors on the global flow.
- From the notation used elsewhere, I think that the text “ $1/n$ as a characteristics time ($t = T/n$)” on line 8 of page 4 should be replaced by “ $1/\omega$ as a characteristics time ($t = T/\omega$)”.
- I found the solutions described by the function $f(T, Y)$ to be slightly confusing, as it is never explicitly stated what system of equations f is solving. It would be better to explicitly give a solution for U in terms of the Newtonian solution V . For example, if the notation was extended to $V(T, Y, \beta)$, then one could write the solution explicitly as $U = (1 - c)^{-1} V(T, Y, \beta_\infty)$.
- In figure 1, I think it is a bit misleading to show the dotted boxes as asymptotic expansions in *only* β . If there is only an expansion in β , then the solution will only be formally valid for $Cu = 0$ or $Cu = \infty$, so the boxes should not extend vertically at all. I think what is probably meant is that each of the dotted boxes is a double expansion in Cu and β .
- On page 12, line 52, I believe that it is incorrect to say that the flow is always of transitional character for $Cu = O(1)$. Specifically, if $Cu = O(1)$ while $\beta \gg 1$ and we are in the Womersley flow regime, then the velocity and shear rate scalings associated with the Womersley flow mean that the effective Carreau number is $Cu/\beta \ll 1$, i.e. we would be in the low shear rate regime.

(This issues arises because the definition of Cu uses the viscous velocity scale and channel width length scale, which are not appropriate for

Womersley flow. The suggested additional regime diagram would help elucidate this point.)

- I am not sure what the paragraph at the bottom of page 16 is supposed to mean. First we are told that something is independent of the other parameters, and then that it is suggested that the other parameters can change things (accelerate or delay the convergence). It cannot be both independent of those parameters and affected by them at the same time. The meaning should be clarified.

Notational and presentational issues

- Is there a reason why the shear-thinning index is n_c rather than the simpler n ?
- The usual symbol for the Womersley number is α , and it would be easier for readers to follow if this convention was used.
- Is there a reason why the viscosity ratio parameter is $1 - \mu_\infty/\mu_0$, rather than the more obvious μ_∞/μ_0 ? The latter would have the advantage that the interesting limit of $\mu_\infty/\mu_0 \rightarrow 0$ would be described by the parameter itself being small.
- In many places pairs of brackets are not large enough to enclose their contents. In \LaTeX , `\left(` (and `\right)`) can be used to ensure appropriate sizing.
- There is a specific symbol \ll for “much less than” (obtained in \LaTeX by `\ll`), rather than using two chevrons ($<<$). Similarly for “much greater than”. (e.g. in the caption for figure 1.)
- There are a few multi-line equations where the horizontal alignment should be improved. For example: page 10 line 20, where the equality/inequality signs should be aligned, and (5.7) where the second line of the right-hand side should start to the right of the equals sign in the line above.

Results Graphs

- For figure 3, I think it would be better to just have a single plot, showing the two Newtonian limits, and then include several different values of Cu from e.g. 0.01 to 10^5 . This would then be comparable to figure 2.
- In figure 4 the lines plotted seem rather under-resolved. Could more points be added to give smoother lines? (The two analytic curves at least should be completely smooth.)
- The graphs in figures 2, 3 and 4 show the three possible transitions as Cu varies at fixed β . It would be helpful if the same style was

adopted for each. i.e. the same two line styles should be used for the two analytic curves for large and small Cu , and the same colours used for the numerical curves at the different values of Cu .

English language and terminology issues

The following list is not exhaustive, but includes some of the more significant and/or repeated mistakes that I spotted.

- “correspondent to” should be “corresponding to”.
- The section heading on page 5 “Dimensional analysis. Basic numbers.” would be more usually written as “Scaling analysis and dimensionless groups”.
- In many places, phrases of the form “when $X \rightarrow 0$ ” (or equivalently “when $X \rightarrow \infty$ ” are used. It would be preferable to be clearer as to whether the limiting value itself is being referred to (e.g. by saying “in the limit $X \rightarrow 0$ ” or “at $X = 0$) or alternatively if the parameter is just meant to be asymptotically close to the limit (e.g. by saying “for $X \ll 1$ ”).
- “independently on” should be “independently of”.
- Instead of “high and small” use either “high and low” or “large and small”.
- On page 5 line 30, instead of “intermediate region” I would use “interior region”. (In boundary-layer theory, the term “intermediate” is usually used for the region between two layers in which the matching is carried out. This is not what is meant here.)
- page 6, line 48: It should be “monotonically” not “monotonously”.
- I would tend to call the results that are obtained in section 4 as “bounds” rather than “estimates”.

Appendix C

We wish to thank both referees for their careful reading of the paper. We are grateful for their remarks that helped us improve the paper. Please find our responses below (the text from the referees is shown in italic fonts and the indications for the text changes - in red color).

Response to the referee 1:

We highly appreciate the referee general comments, which are fundamental concerning the studied problem:

A *I believe that the ability to place bounds on such flows is potentially useful, although my principal concern is the very limited regime under which these results apply, i.e. perfectly straight 2D channels. I appreciate the analysis here seems to be highly reliant on a unidirectional flow, but I wonder whether considering the influence of asymptotically-small wall perturbations might expand the study's applicability while still leaving it accessible to the kind of analysis being applied? In its current limited form, I just wonder whether it sufficiently advances our understanding of these kinds of flows through channels.*

We agree with the referee that the flows in perfectly straight 2D channels are with limited regime applications.

The flows in perfectly straight 2D channels are considered in the present work. In fact, small disturbances of the channel width can be added to the model. Then the dimensionless wall position will be given as $Y = -\varepsilon g(T)$ and $Y = 1 + \varepsilon g(T)$, where $|g(T)| \leq 1$ and $\varepsilon \ll 1$. In this way, the present problem will occur the zero-th order approximation in ε of the more general problem of wall perturbations in time (from elastic or other sources). For example, the proper knowledge of the flow velocity in rigid channels/tubes is a starting point before the introduction of elasticity in the model, as a fluid-structure interaction.

B *This is perhaps also compounded by the final conjecture concerning the Carreau number being solely responsible for the transition of the flow between the two Newtonian regimes. Unless I am missing something, is this not immediately obvious from the non-dimensional form of (2.1)?*

We thank the referee for this suggestion. The first impression of eq. (2.1) seems like that the transition between the two Newtonian regimes depends only on the Carreau number. However, the expression $\lambda\dot{\gamma}$, or its dimensionless correspondent $Cu\frac{\partial U}{\partial Y}$, is not under control (in mathematical sense), i.e., it is not clear what will happen when $Cu \rightarrow \infty$ and $\frac{\partial U}{\partial Y} \rightarrow 0$, which is the case of some polymers (Cf. R.I. Tanner, Engineering rheology, Oxford ed., 2002.) The latter case is discussed in Chapter 5b of the present paper at $\beta \gg 1$ and $\frac{Cu}{\beta} \gg 1$

Below we list successively the answers to the referee other comments, as follows:

- 1 *The authors appear to consider a general time dependence, but when comparing between theory and numerical simulations allow transients to decay leaving only purely oscillatory motion. However, there are plenty of applications where transients are of interest (including in blood flow), and so if the bounds are valid both for oscillatory and non-oscillatory transient flows this could be worth exploring more.*

We thank the referee also for this suggestion.

Another further continuation of the present work is to use a more general function $G(T)$ of the pressure gradient instead of the pure oscillation in eq. (2.5). $G(T)$ must be bounded and smooth enough. The obtained bounds for the Carreau velocity will be similar, but it is necessary to know explicitly the function $G(T)$. Moreover, for general function $G(T)$, the solution of the Newtonian velocity can not be given in a closed analytic form like in the present work for $G(T) = \cos T$.

- 2 *On page 4, line 8: Should $1/n$ read $1/\omega$, given the definition of beta below?*

Corrected.

- 3 *There were some statements on p14, the logic of which I was not able to completely follow. For example, why does $\Phi(dU_\infty/dy)$ taking values between 0 and 1 combined with Lemma 4.2 mean that it governs the solution? Also, why does $\Phi(dV_\infty/dy)$ being differently slightly from 1*

means that U_∞ can be approximated by V_∞ up to the stated order of accuracy? Some clarification on these statements would be useful.

The idea is to show for what range of the parameters Cu and β , the solution U_∞ is close to the Newtonian solution V_∞ . For this purpose eq. (5.5) is rewritten as eq. (4.7), i.e.:

$$8 \frac{\partial U_\infty}{\partial T} - \frac{1}{\beta^2} \frac{\partial}{\partial Y} \left\{ \left[1 - c + c \Phi \left(\frac{\partial U_\infty}{\partial Y} \right) \right] \frac{\partial U_\infty}{\partial Y} \right\} - \frac{1}{\beta^2} \cos(T) = 0, \quad (1)$$

Then, if we substitute the solution U_∞ by the first term of its asymptotic expansion eq. (5.6), $U_{\infty 0}$, the function $\Phi \left(\frac{\partial U_{\infty 0}}{\partial Y} \right)$ will differ slightly from 1 for a large range of $\frac{Cu}{\beta} \ll 1$. The latter follows from Lemma 4.2., where $\eta \sim O \left(\frac{1}{\beta} \right)$ and $1 - \Phi(\eta) \leq \min \left\{ 1, \frac{3}{2}(1-n)Cu^2\eta^2 \right\} \approx 0$ at $\frac{Cu}{\beta} \ll 1$.

If the solution U_∞ is substituted by $U_{\infty 0}$ in Φ , then $\Phi \left(\frac{\partial U_{\infty 0}}{\partial Y} \right)$ differs slightly from 1 for a large range of $\frac{Cu}{\beta} \ll 1$. This means that the solution $U_\infty(T, Y)$ can be approximated by $V_{\infty 0}(T, Y)$ up to $O \left(\frac{1}{\beta^4} \right)$. Thus, it occurs that, if $\frac{Cu}{\beta} \ll 1$, the solution $U_{\infty 0} \approx V_{\infty 0}$.

- 4 *Why in Figure 2 is this very particular value of beta considered, it seems curious that the number has been specified up to 4 decimal places?*

The value 0.8839 of β corresponds to the blood flow in common carotid artery of diameter 6.65mm [24].

- 5 *It might be interesting to have K_1^0 and K_1^1 plotted on the same axes as a function of c , to gauge the values of c and which these diverge.*

From the expressions for K_1^0 and K_1^1 given by eqs. (4.15) and (4.19), it is obvious that $K_1^0 \geq K_1^1$ for all values of c , Cu and n . Moreover, K_1^1 tends to K_1^0 at $n \rightarrow 0$ and/or $Cu \rightarrow \infty$. Here, we include a plot to

Fig.2 Plots of K_1^0 and K_1^1 as functions of c for different n from 0.1 to 0.9 by 0.2 and $Cu = 1$.

show this tendency given in Fig.2.

- 6 *I am not sure I am convinced by the case for the flow remaining laminar, since the critical Reynolds number cited from Patel and Head does not appear to come from non-Newtonian flow experiments*

We expect that the critical Reynolds number, Re_{cr} , of Carreau fluid flows in straight channels will not be very different from 1300. Although that we haven't found any experimental confirmation for Re_{cr} of shear-thinning flows in channels, there are many results for circular pipes, that support this idea. For example, the experiments show that, the transition to turbulence of shear-thinning flows in pipes may be delayed in comparison to Newtonian fluids [28].

[28] Pinho FT, Whitelaw JH. 1990. Flow of non-Newtonian fluids in a pipe. *Journal of Non-Newtonian Fluid Mechanics* **34**, Issue 2, 129–144.

Response to the referee 2:

We wish to thank the referee for his/her positive appreciation of our paper as having potential to be published. We agree with the referee that the paper needs more explanation concerning the representation in the parameter map. We tried to improve the text to be better understood.

Layout, structure and narrative issues

- 1 *The manuscript would be much improved if it were clearer in the abstract and introduction exactly what it was trying to achieve. I would also suggest that it would be helpful if section 5 was split into two for the two different limits, and if both of these sections came before section 4.*

As suggested by the referee, Section 5 is split into two subsections: 5a) $\beta \rightarrow 0$ and 5b) $\beta \rightarrow \infty$. However, it is not possible to transpose Section 5 before Section 4, since some of the bounds found in section 4 are later used in Section 5.

- 2 *I think figure 1 should be shown and described much sooner, to help readers understand the parameter space. In addition, I would like to see another similar figure, showing the different flow regimes across the same parameter space. The flow regime can be categorised in two ways: First, the flow profile can either be viscous-dominated Poiseuille flow, inertia-dominated Womersley flow, or somewhere between the two. Secondly, the viscosity can either be dominated by μ_0 for low shear, be dominated by μ_∞ for high shear, or be a mixture of the two. Showing the regions occupied by these different regimes on a parameter-space diagram would be most helpful.*

We thank the referee for this proposition. A new figure similar to Fig.1 (in the old version) is created and inserted in the Introduction. This new figure is denoted as Fig.1.

(usually named Womersley or inertia flow regime). The position of the Poiseuille flow, Womersley flow and flow between these two

regime flows is given in Fig.1 as a parameter-space diagram of Carreau number and Womersley number. Also, the diagram contains the position of low shear, corresponding to high viscosity, and high shear - low viscosity.

Fig.1 Sketch of the different flow regimes in the parameter-space diagram of Womersley number β and Carreau number Cu .

3 *In terms of following the work and understanding what is being done and why, I think it would be very helpful to readers if it were explicitly stated (if I have have understood things correctly) that the objective is to obtain bounds on the velocity and velocity gradient; either on their absolute magnitude, or on the deviation from a comparable Newtonian solution. Setting this out more clearly at the beginning would have helped me greatly when reading the heavy-going section 4 of the manuscript.*

We tried to better explain the objective of this work in the abstract:

It is shown that the Carreau number changes the type of the flow velocity to be closer to the Newtonian velocity corresponding to low or high shear or to have a transitional character between both Newtonian velocities. Some numerical examples for the velocity at different Carreau and Womersley numbers are presented for illustration with respect to the similar Newtonian flow velocity.

and in the Introduction:

(usually named Womersley or inertia flow regime). The position of the Poiseuille flow, Womersley flow and flow between these two regime flows is given in Fig.1 as a parameter-space diagram of Carreau number and Womersley number. Also, the diagram contains the position of low shear, corresponding to high viscosity, and high shear - low viscosity.

The proven bounds for the Carreau velocity, its gradients and the bounds for the absolute difference between the Newtonian and Carreau velocity solutions will be shown to be valid for every Womersley number, Carreau number and rheological power coefficient n . However, the bound for the absolute difference between the Newtonian and Carreau velocity solutions will occur more useful at low values of Carreau number or in the limit $n \rightarrow 1$. At high Womersley numbers, it will be shown that the effective Carreau number is responsible for solution type, i.e., if the solution can be approximated with one or the other Newtonian velocity corresponding to low or high shear viscosity or will have a transitional character.

- 4 *It would also be good if the authors could say a bit more about why these bounds are useful / where they might be used, and also how tight they are. Could a summary be provided of which of the bounds are most useful in which regions of parameter space?*

The proven bounds for the Carreau velocity and its gradients, given by K_1 on eqs. (4.29) and (4.32) and the bounds for the absolute difference between the Newtonian and Carreau velocity

solutions - by K_2 , eqs. (4.33), (4.34) are valid for every $\beta \in (0, \infty)$, $n \in (0, 1)$ and $Cu \in (0, \infty)$. However, the bound K_2 , eqs.(4.33) and (4.34), is more useful at $Cu \ll 1$ or in the limit $n \rightarrow 1$.

Technical and scientific issues

- 1 *I do not think it is completely obvious that the $Cu \gg 1$ solutions necessarily tend to the Newtonian solution with viscosity μ_∞ (e.g. page 5, line 41). Is it not possible that the small regions where the shear rate is small have a global effect on the flow profile? (It should be noted that the spatial extent of these regions is time-dependent, and for short periods of time in each cycle the spatial regions will not be small.) I think more should be said here, perhaps with an order of magnitude estimate (in terms of Cu) for the errors on the global flow.*

It is true that at $Cu \gg 1$ the solutions do not necessarily tend to the Newtonian solution with viscosity μ_∞ . It is difficult to define precisely the regions of low shear, where the two solutions differ, since they are time-dependent. Moreover, for high values of Cu , it is not possible to prove exactly the order of magnitude of the Carreau solution or the difference between both solutions in terms of Cu .

In the general case of a Carreau fluid, at $Cu \neq 0$, the velocity satisfying eqs. (2.5) and (2.6), can be found only numerically. In the limiting case of low $Cu \rightarrow 0$, there exists an asymptotic solution, given in [15]. For $Cu/\beta \rightarrow \infty$ the solution of eqs. (2.5) and (2.6) is found numerically to be close to the solution $\frac{f(T, Y)}{1 - c}$.

- 2 *From the notation used elsewhere, I think that the text "1/n as a characteristics time ($t = T/n$)" on line 8 of page 4 should be replaced by "1/ ω as a characteristics time ($t = T/\omega$)".*

Corrected.

- 3 *I found the solutions described by the function $f(T, Y)$ to be slightly confusing, as it is never explicitly stated what system of equations f is solving. It would be better to explicitly give a solution for U in terms of the Newtonian solution V . For example, if the notation was extended to $V(T, Y, \beta_\infty)$, then one could write the solution explicitly as $U = (1c)^{-1}V(T, Y, \beta_\infty)$.*

We include eq. (3.4) (according to the revised version of the paper) for the function $f(T, Y)$.

If the lower viscosity μ_∞ is used as a characteristic viscosity and $B_\infty = \frac{AH^2}{\mu_\infty} = \frac{B}{1-c}$ as characteristic velocity, the corresponding Newtonian flow velocity $f(T, Y)$ satisfies the equation:

$$8\beta_\infty^2 \frac{\partial f}{\partial T} - \frac{\partial^2 f}{\partial Y^2} - \cos(T) = 0,$$

where $\beta_\infty = \beta \sqrt{\frac{\mu_0}{\mu_\infty}} = \frac{\beta}{\sqrt{1-c}} > \beta$. The solution $f(T, Y)$ has the same form as (3.1) – (3.3) at β replaced by β_∞ [14].

- 4 *In figure 1, I think it is a bit misleading to show the dotted boxes as asymptotic expansions in only β . If there is only an expansion in β , then the solution will only be formally valid for $Cu = 0$ or $Cu = \infty$ so the boxes should not extend vertically at all. I think what is probably meant is that each of the dotted boxes is a double expansion in Cu and β*

Fig. 1 of the old version, in the reviewed version denoted as Fig. 3, is elaborated and better exposed.

- 5 *On page 12, line 52, I believe that it is incorrect to say that the flow is always of transitional character for $Cu = O(1)$. Specifically, if $Cu = O(1)$ while $\beta \gg 1$ and we are in the Womersley flow regime, then the velocity and shear rate scalings associated with the Womersley flow mean that the effective Carreau number*

Fig.3. Different regimes for the velocity solution $U(T, Y)$ with respect to Womersley number β and Carreau number Cu : (I) $Cu \ll 1$ (low shear viscosity region) - asymptotic expansion in Cu^2 [15]: (I.1) $\beta \ll 1$: $U_0 = U_{00} + O(\beta^2) + O(Cu^2)$, where $U_{00} \rightarrow V_{00}$; (I.2) $\beta \sim O(1)$: $U = V + O(Cu^2)$; (I.3) $\beta \gg 1$: $U_\infty = U_{\infty 0}/8\beta^2 + O(1/\beta^4) + O(Cu^2)$, where $U_{\infty 0} \rightarrow V_{\infty 0}$; (II) $Cu \sim O(1)$ (transitional shear viscosity region): (II.1) $\beta \ll 1$: $U_0 = U_{00} + O(\beta^2)$, where U_{00} - numerical solution; (II.2) $\beta \sim O(1)$ - numerical solution; (II.3) $\beta \gg 1$ and $Cu/\beta \ll 1$: $U_\infty = U_{\infty 0}/8\beta^2 + O(1/\beta^4)$, where $U_{\infty 0} \rightarrow V_{\infty 0}$; (III) $Cu \gg 1$ (high shear viscosity region): (III.1) $\beta_\infty \ll 1$: $U_0 = U_{00} + O(\beta^2)$, where $U_{00} \rightarrow f_{00}/(1-c)$; (III.2) $\beta_\infty \sim O(1)$ - numerical solution; (III.3) $\beta \gg 1$ and $Cu/\beta \sim O(1)$: $U_\infty = U_{\infty 0}/8\beta^2 + O(1/\beta^4)$, where $U_{\infty 0}$ - numerical solution; (III.4) $\beta \gg 1$ and $Cu/\beta \gg 1$: $U_\infty = U_{\infty 0}/8\beta^2 + O(1/\beta^4)$, where $U_{\infty 0} \rightarrow f_{\infty 0}$. The red lines are as in Fig.1

is $Cu/\beta \ll 1$, i.e. we would be in the low shear rate regime. (This issues arises because the definition of Cu uses the viscous velocity scale and channel width length scale, which are not appropriate for Womersley flow. The suggested additional regime diagram would help elucidate this point

The referee is right. We have revised the text on p. 12 (old version) and replaced with the following (on p. 13 of revised version):

For $Cu \gg 1$ and $\frac{Cu}{\beta} \gg 1$, the solution in the boundary layer can be found only numerically or by some approximate methods. At $\frac{Cu}{\beta} \gg 1$, the solution $U_{\infty 0}$ tends numerically to $f_{\infty 0}(T, Y)$, which is the first term in (3.10), i.e.:

$$f_{\infty 0}(T, Y) = \sin T - \exp\left(\frac{-2\beta}{\sqrt{1-c}}Y\right) \sin\left(T - \frac{2\beta}{\sqrt{1-c}}Y\right) - \exp\left(\frac{-2\beta}{\sqrt{1-c}}(1-Y)\right) \sin\left(T - \frac{2\beta}{\sqrt{1-c}}(1-Y)\right). \quad (2)$$

In this case the boundary layers are thinner (with width $O(\frac{\sqrt{1-c}}{\beta})$) than those of the Newtonian flow.

In Fig.3 we present a map of Womersley vs Carreau number space, in loglog scale for the different approximations of the velocity $U(T, Y)$. From Fig.3 it is seen the transitional character of $Cu = O(1)$ (in sense of transition between the two Newtonian solutions corresponding to low and high shear), which concerns the Poiseuille and transition flow regime (the flow regime between the viscous Poiseuille flow and inertia Womersley flow). In the Womersley flow regime, the transition between the two Newtonian solutions is given by the effective Carreau number $Cu/\beta = O(1)$.

- 6 *I am not sure what the paragraph at the bottom of page 16 is supposed to mean. First we are told that something is independent of the other parameters, and then that it is suggested that the other parameters can change things (accelerate or delay the convergence). It cannot be both independent of those parameters*

and affected by them at the same time. The meaning should be clarified.

This paragraph is rewritten in order to clarify that the effective Carreau number is Cu in the Poiseuille and transition regime and Cu/β in the Womersley flow regime.

Finally, we could make the following statement, that the effective Carreau number is Cu in the Poiseuille and transition regime and Cu/β in the Womersley flow regime. This leads to the conjecture: basically the effective Carreau number, is responsible for solution type changes, converging to one or the other Newtonian solution. The other parameters: Womersley number β , n and c can only accelerate or delay this convergence process when increasing the effective Carreau number.

Notational and presentational issues

- 1 *Is there a reason why the shear-thinning index is n_c rather than the simpler n ?*

Corrected.

- 2 *The usual symbol for the Womersley number is α , and it would be easier for readers to follow if this convention was used.*

Usually in the literature the symbol for Womersley number is $\alpha = H\sqrt{\frac{\rho\omega}{\mu_0}}$. Here, we use $\beta = \frac{\sqrt{2}}{4}\alpha$ for convenience with the presentations in [14] and [15].

- 3 *Is there a reason why the viscosity ratio parameter is $1 - \mu_\infty/\mu_0$, rather than the more obvious μ_∞/μ_0 ? The latter would have the advantage that the interesting limit of $\mu_\infty/\mu_0 \rightarrow 0$ would be described by the parameter itself being small.*

Here, we use $c = 1 - \mu_\infty/\mu_0$ for convenience with the presentation in [15].

- 4 *In many places pairs of brackets are not large enough to enclose their contents. In LATEX, (and) can be used to ensure appropriate sizing.*

Corrected.

- 5 *There is a specific symbol for "much less than" (obtained in LATEX by \ll), rather than using two chevrons ($\langle\langle$). Similarly for "much greater than". (e.g. in the caption for figure 1.)*

Corrected.

- 6 *There are a few multi-line equations where the horizontal alignment should be improved. For example: page 10 line 20, where the equality/inequality signs should be aligned, and (5.7) where the second line of the right-hand side should start to the right of the equals sign in the line above.*

Corrected.

Results Graphs

- 1 *For figure 3, I think it would be better to just have a single plot, showing the two Newtonian limits, and then include several different values of Cu from e.g. 0.01 to 10^5 . This would then be comparable to figure 2.*

The figures 3a) and 3b) are combined and replaced by the single Fig.5 (revised version).

- 2 *In figure 4 the lines plotted seem rather under-resolved. Could more points be added to give smoother lines? (The two analytic*

curves at least should be completely smooth.)

The analytical curves in Fig.6 (revised version) are replaced by smoother ones.

- 3 *The graphs in figures 2, 3 and 4 show the three possible transitions as Cu varies at fixed β . It would be helpful if the same style was adopted for each. i.e. the same two lines styles should be used for the two analytic curves for large and small Cu , and the same colours used for the numerical curves at the different values of Cu .*

The styles and colors of the graphs are unified.

English language and terminology issues

- 1 *"correspondent to" should be "corresponding to"*

Corrected.

- 2 *The section heading on page 5 "Dimensional analysis. Basic numbers." would be more usually written as "Scaling analysis and dimensionless groups".*

Changed.

- 3 *In many places, phrases of the form "when $X \rightarrow 0$ " (or equivalently "when $X \rightarrow \infty$ " are used. It would be preferable to be clearer as to whether the limiting value itself is being referred to (e.g. by saying "in the limit $X \rightarrow 0$ " or "at $X = 0$) or alternatively if the parameter is just meant to be asymptotically close to the limit (e.g. by saying "for $X \ll 1$ ").*

Changed appropriately.

4 "independently on" should be "independently of"

Changed.

5 *Instead of "high and small" use either "high and low" or "large and small".*

Changed to "high and low".

6 *On page 5 line 30, instead of "intermediate region" I would use "interior region". (In boundary-layer theory, the term "intermediate" is usually used for the region between two layers in which the matching is carried out. This is not what is meant here.)*

Changed.

7 *page 6, line 48: It should be "monotonically" not "monotonously".*

Changed.

8 *I would tend to call the results that are obtained in section 4 as "bounds" rather than "estimates".*

Changed appropriately.

Appendix D

We are grateful to the referees for their valuable remarks that helped us to improve the paper. Our responses are given below in bold, the text from the referees - in italic and the changes in the paper text, where applicable, - in red.

Response to the referee 1:

- 1 *I still feel as though the authors have somewhat missed an opportunity in not applying their analysis to non-oscillatory flows, even if it requires a partial numerical treatment. It would certainly help to move the applicability of the analysis closely to the stated area of interest, i.e. blood flow, which of course is not perfectly oscillatory.*

We thank the referee for this remark. In fact, the case of an arbitrary pressure gradient function of time is under research and will be included in our subsequent work. Although that, there is no analytic solution for the Newtonian flow case, our preliminary conclusions show that the theoretical estimates will not be very different from the presented here, since we do not use the analytic form of the Newtonian solution explicitly.

- 2 *I note the authors' response as to whether or not the conclusions about the Carreau number controlling the transition are obvious, but feel as though their arguments perhaps need fleshing out a little more. In particular, providing greater insights into situations when dU/dY would tend to zero in the presence of no-slip boundaries.*

It is possible for some times T , the gradient $\frac{\partial U}{\partial Y} = 0$ on the no-slip boundaries $Y = 0$ and $Y = 1$, which follows from the continuity of the solution $U(T, Y)$ and its gradient with respect to time T and space coordinate Y .

- 3 *I also would have thought that a more interesting case would be when Cu tends to zero but dU/dY tends to infinity. However, I am guessing*

that for the straight channel problem neither of these cases are likely, and so in this context it still seems as though the conclusions are somewhat obvious for the problem being solved. I will not labour this point further, but the authors may like to reflect more about these more interesting scenarios in their Conclusion sections.

We agree with the referee that the case when Cu tends to zero but dU/dY tends to infinity is an interesting one. In connection with this, we add the following comment in the Conclusion:

Finally, we would like to point out the open problem connected with the special case of $c = 1$. Then the equation (2.5) is not uniformly parabolic one and the existence of a classical solution is questionable. In this case it is possible the gradient $\frac{\partial U}{\partial Y}$ on the boundaries $Y = 0$ and $Y = 1$ to become infinite for some times T . However, for $c \in [0, 1)$, the gradient $\frac{\partial U}{\partial Y}$ can not reach infinite values nowhere inside the region $Y \in [0, 1]$ and $T \geq 0$ according to Corollary 4.1. Our conjecture is that at $c = 1$ and $Cu \rightarrow 0$, the gradient is bounded and the treated problem still has a classical solution.

- 4 *With regards the value of beta, and would still encourage them to reflect on whether 4 decimal places is entirely appropriate given the idealisations already in their model.*

Corrected up to 3 decimal digits.

Response to the referee 3:

- 1 *The authors derive bounds for velocity gradients and velocity itself, by approximation to solutions of an oscillatory flow in a planar channel due to an imposed sinusoidal pressure gradient, for a fluid whose viscosity follows the Carreau model. For that, they expand the (unknown) solution in terms of Beta (the so-called Womersley number, which is the square-root of the dimensionless frequency), when Beta is small, or $1/\text{Beta}$ when Beta is large. This, for me, is a strange procedure because the basic steady solution for Poiseuille flow (for $\text{Beta}=0$) is unknown for the Carreau model. So in the perturbation method utilized, when the governing equation (or the "expected" solution) is expanded in a series in terms of the small parameter Beta, the first element of the series is not known.*

At $\beta = 0$, eq. (5.2) for U_{00} becomes quasi-stationary, whose solution is unknown, as it depends on the Carreau number. According to the analysis shown in II.1 of Fig.3, the solution U_{00} can be found only numerically, which is given in the caption and also added now in the Fig.3 itself.

- 2 *The corresponding procedure for the Newtonian case is well illustrated in White's book (see below), showing the occurrence of the Richardson's annular effect (similar to the velocity profiles given here in Figs 4 and 6). I am surprised to notice that even these classical references are not cited.*

The referee is right, this problem for Newtonian fluids is well known and discussed in many classical books. The Richardson annular effect concerns pipe flow, which is not very different from the channel flow. The following text is included in the Introduction:

The pipe flow of Newtonian fluid due to oscillating pressure gradient has been first studied experimentally by Richardson and Tyler [13]. Their observations of the maximum velocity displacement towards to the wall is known as the Richardson's annular effect, which is explained

also using the analytical solution for the velocity [14], [15]. The channel flow has similar behaviour with a little different analytical solution as found in [16] and later used by different authors to validate their numerical solutions for non-Newtonian fluid flows, e.g., [17]–[19].

- 3 *One is tempted to ask the authors why haven't they tried the same problem with, for example, the power-law viscosity model? At least they would know the solution for $\text{Beta}=0$. (I have not check the literature to see if that problem has already been considered by other authors).*

The choice of the Carreau viscosity model is connected with our recent interests in problems of blood flows in arteries, polymer flows during 3D printing, etc., where the high shear viscosity $\mu_\infty > 0$ and the Carreau model is usually applied.

- 4 *In addition, I think the authors have not done an adequate literature review. The solution for the Newtonian fluid is a classical solution in fluid mechanics (or mathematical physics), given for example in the book of White (Viscous Fluid Flow). However the authors present it (Eqs. 3.1-3.3) as if it was derived by themselves. Many other authors have used that solution, for example Duarte et al (J Non-Newt Fluid Mech. 154 (2008) 153-), who also considered the pulsating flow of viscoelastic fluids, and in particular with the Carreau model, Miranda et al (Int J Num Meth Fluids 57 (2008) 295). As noted above, the classical papers, such as Sxrl (1930) (Z Phy 61, 349-, first solution for pipe flow), Richardson & Tyler (1929) (Proc Phy Soc London, 42, p. 1-15, found the velocity overshoot near wall at large frequencies), are not given.*

We thank the referee for this remark. The citations of these references are appropriately added in the Introduction (the text is given after the answer to remark 2):

[13] Richardson EG, Tyler E. 1929. The transverse velocity gradient near the mouths of pipes in which an alternating or continuous flow of air is established. *Proc. Phys. Soc.* **42**, Part I, No.231, 1–15.

- [14] Schlichting H, Gersten K. 2000. *Boundary Layer Theory*. 8th edition. Springer, Berlin
- [15] White FM. 2006. *Fluid Mechanics*. 3-rd edition. McGraw-Hill, Boston.
- [16] Landau LD, Lifshitz EM. 1959. *Fluid Mechanics*. 1st edition. Pergamon Press, Oxford, London.
- [17] Boyd J, Buick JM, Green S. 2007. Analysis of the Casson and Carreau-Yasuda non-Newtonian blood models in steady and oscillatory flows using the lattice Boltzmann method. *Physics of Fluids* **19**, paper 093103, 14p.
- [18] Miranda AIP, Oliveira PJ, Pinho FT. 2008. Steady and unsteady laminar flows of Newtonian and generalized Newtonian fluids in a planar T-junction. *Int. J. Numer. Meth. Fluids*. **57**, 295–328.
- [19] Duarte ASR, Miranda AIP, Oliveira PJ. 2008. Numerical and analytical modelling of unsteady viscoelastic flows: The start-up and pulsating test case problems, *Journal of Non-Newtonian Fluid Mechanics*. **154**, Issues 2-3, 153–169.

The solution (3.1)-(3.3) is well known, as we mentioned in our previous work Kutev et al, 2015 [20], where cited Boyd et al, 2007 [17]. Here we decided to cite the book of Landau and Lifshitz, 1959 [16], where the channel flow solution is given for the first time, up to our knowledge. The following text is included in Chapter 3:

$V(T, Y)$, is found explicitly for the first time in the classical book of Landau and Lifshitz [16] and also used in our previous papers [20] – [25].

- 5 *This problem has too many dimensionless numbers (the Womersley number, the Carreau number Cu - which has little physical meaning, a viscosity ratio here called c , the power law index n) and so in a future study I suggest the authors concentrate in typical values of these parameter, eg. $N=0.5$, $c=0.99$ or 0.9 , Cu of order 10, 100, and seek the effect of the imposed frequency. It should be aid that obtaining a numerical solution for this problem is straightforward and so the expected*

trends at high/low Beta, or Cu, are easy to find.

This is a good suggestion and we shall take it into account in our future studies on the problem.

- 6 *The values of the Carreau number here employed are unrealistic (for example in Table 1 and Figs. 4,5,6). The authors have failed to recognize that Cu is not independent of n and c. If $Cu = \lambda * \dot{\gamma}$ ($\lambda =$ time constant of Carreau model; $\dot{\gamma} =$ typical shear rate of the flow) is smaller than 1 or greater than a certain limit $Cu_{\infty} = \lambda * \dot{\gamma}_{\infty}$, they simply have a Newtonian fluid with viscosity η_0 or η_{∞} , respectively. This explains their findings at high and low Beta's and Cu's. From simple calculations I get $\log(Cu_{\infty}) = \log(1 - c)/(n - 1)$. For $c = 0.9$, $n = 0.5$ this gives $Cu_{\infty} = 100$. Using Cu greater than this value gives results for a Newtonian fluid with viscosity = η_{∞}*

Although that the referee finds the higher values of Carreau number in Figs. 4,5,6 and Table 1, we think that such values may occur interesting for some special cases.

Minor points

- 7 *Page 3, calling lambda the relaxation time is misleading; lambda is just a constant of the model with units of time. 1/lambda is the shear rate at which the viscosity starts to decrease.*

Corrected.

- 8 *Page 4, Eqs 2.2 and 2.3 are not necessary; suffices to give the momentum equation that is actually solved (a much simpler version of the Navier-Stokes equations)*

We prefer to leave these equations to make clearer the statement for non-specialists.

9 *Page 4, Eq. 2.4 is wrong (this is probably just a mistype)*

Corrected

10 *Fig. 4, why the need to specify the time $T=20*\pi$? The flow should be sinusoidal (stationary in time) and the exact time moment is irrelevant (except inside a cycle)*

Corrected.